TOOLS

**JCB** Journal of Cell Biology

# Optogenetic and chemical genetic tools for rapid repositioning of vimentin intermediate filaments

Milena Pasolli[1]*, Joyce C.M. Meiring[1]*, James P. Conboy[2], Gijsje H. Koenderink[2], and Anna Akhmanova[1]

**Intermediate filaments (IFs) are a key component of the cytoskeleton, essential for regulating cell mechanics, maintaining nuclear integrity, organelle positioning, and modulating cell signaling. Current insights into IF function primarily come from studies using long-term perturbations, such as protein depletion or mutation. Here, we present tools that allow rapid manipulation of vimentin IFs in the whole cytoplasm or within specific subcellular regions by inducibly coupling them to microtubule motors, either pharmacologically or using light. Rapid perinuclear clustering of vimentin had no major immediate effects on the actin or microtubule organization, cell spreading, or focal adhesion number, but it reduced cell stiffness. Mitochondria and endoplasmic reticulum (ER) sheets were reorganized due to vimentin clustering, whereas lysosomes were only briefly displaced and rapidly regained their normal distribution. Keratin moved along with vimentin in some cell lines but remained intact in others. Our tools help to study the immediate and local effects of vimentin perturbation and identify direct links of vimentin to other cellular structures.**

## Introduction

Intermediate filaments (IFs), along with actin and microtubules, are among the three major cytoskeletal systems in mammalian cells (Herrmann and Aebi, 2004). Unlike microtubules and actin, IFs lack polarity, exhibit much slower dynamics, and are extremely resilient and elastic (Etienne-Manneville, 2018; Herrmann et al., 2007). Among them, vimentin, a type III IF protein, is a prominent component of cells of mesenchymal origin (Mendez et al., 2010). Recent structural work indicated that vimentin forms a helical structure of 40 α-helices organized into five protofibrils that are connected by the vimentin tails, with the head domains located in the lumen (Eibauer et al., 2024). Vimentin IFs surround the nucleus and spread throughout the cytoplasm, a distribution facilitated by microtubule-based motors (Hookway et al., 2015; Robert et al., 2019). Additionally, due to interactions with actin filaments, the vimentin network is subject to actin-driven retrograde flow (Leduc and Etienne-Manneville, 2017).

Vimentin plays an important role in many cellular processes, such as cell mechanics and control of cell stiffness (Guo et al., 2013; Pogoda and Janmey, 2023), maintenance of cell shape, cell-substrate adhesion, and cell migration (Bhattacharya et al., 2009; Cheng and Eriksson, 2017; Eckes et al., 2000; Eckes et al., 1998; Venu et al., 2022, Preprint). Furthermore, vimentin participates in templating and stabilizing microtubule networks (Gan et al., 2016; Schaedel et al., 2021) and is involved in the

positioning and anchorage of organelles such as mitochondria (Nekrasova et al., 2011), endo-lysosomes, and endoplasmic reticulum (ER) (Cremer et al., 2023). Finally, vimentin plays a role in signaling and gene regulation (Paulin et al., 2022).

The cellular functions of vimentin have been primarily studied using knockout and knockdown approaches (Bhattacharya et al., 2009; Colucci-Guyon et al., 1994; Eckes et al., 2000; Kural et al., 2007; Mendez et al., 2014; Mohanasundaram et al., 2022; Ostrowska-Podhorodecka et al., 2022). While these studies have provided insights into the consequences of vimentin loss, they could not distinguish the direct and immediate interactions and interdependencies between vimentin and other cellular structures from the long-term effects caused by the absence of vimentin and the ensuing compensatory cellular changes, including alterations in gene expression.

Treatments perturbing vimentin organization have also been used to study its function (Ridge et al., 2016). For example, overexpression of defective gigaxonin, an E3 ligase adaptor, induces dense perinuclear aggregation of vimentin filaments (Bomont, 2016; Mahammad et al., 2013). Similarly, withaferin A, a natural product with anti-tumor and antiangiogenesis properties, reorganizes vimentin into perinuclear aggregates but lacks specificity, impacting not only other IFs but also actin and microtubules (Bargagna-Mohan et al., 2007; Grin et al., 2012).

[1]Cell Biology, Neurobiology and Biophysics, Department of Biology, Faculty of Science, Utrecht University, Utrecht, The Netherlands; [2]Department of Bionanoscience, Kavli Institute of Nanoscience Delft, Delft University of Technology, HZ Delft, The Netherlands.

*M. Pasolli and J.C.M. Meiring contributed equally to this paper. Correspondence to Anna Akhmanova: a.akhmanova@uu.nl.

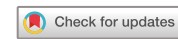

Another pharmacological agent, simvastatin, was shown to induce bundling and collapse of vimentin filaments (Lavenus et al., 2020; Trogden et al., 2018), but its use as a tool is complicated by additional effects, such as its impact on cholesterol synthesis. Furthermore, microinjected mimetic peptides were used to disrupt filament assembly and rapidly reorganize the vimentin network (Goldman et al., 1996; Helfand et al., 2011). While these peptides act quickly—causing changes within minutes—they often disrupt other cytoskeletal components, leading to cellular alterations like rounding and blebbing (Goldman et al., 1996). Their specificity is further limited as they are derived from conserved IF regions (Strelkov et al., 2002), potentially affecting other IF networks. Additionally, microinjection requires specialized expertise and equipment, reducing accessibility and reproducibility. Altogether, this underlines the need for more precise tools to reorganize vimentin IFs and directly observe their interactions and effects on other cellular structures.

In this study, we developed a robust method to rapidly relocalize vimentin filaments, either locally or globally, by recruiting to them microtubule-based motors using either chemical or light-induced heterodimerization. This allowed us to observe vimentin interactions and interdependencies with other cellular structures and their dynamic behavior on a time scale of 15–60 min, which is too short to alter gene expression. We found that clustering of vimentin in the cell center by microtubule minus end–directed kinesins had only mild effects on the actin and microtubule cytoskeleton and no major impact on cell spreading and focal adhesion number but strongly reduced cell stiffness. The keratin-8 network was not affected by vimentin displacement in HeLa cells but was co-clustered with vimentin in U2OS and COS-7 cell lines, supporting the cell type–specific nature of IF organization. ER sheets and mitochondria were relocalized together with vimentin, confirming their direct interactions. In contrast, lysosomes were only mildly affected, and their normal distribution was rapidly restored. Our method enables acute and controlled studies of the role of vimentin in cytoskeletal cross-talk and organelle positioning and dynamics, revealing short-term effects and steady-state alterations.

## Results

### Chemically induced repositioning of vimentin to microtubule plus or minus ends

To induce rapid repositioning of vimentin IFs, we used a chemically inducible heterodimerization system to recruit microtubule-based motors to vimentin, as described previously for membrane organelles (Kapitein et al., 2010). In cells with a radial microtubule organization, recruitment of minus end–directed motors concentrates cargoes near the cell nucleus, whereas plus end–directed motors disperse cargoes toward the cell periphery (Kapitein et al., 2010).

The system we used was based on inducible binding of two protein domains, FRB and FKBP, upon the addition of a rapamycin analog (rapalog AP21967, also known as A/C heterodimerizer) (Clackson et al., 1998; Pollock et al., 2000). We fused the FKBP domain and a fluorescent marker, mCherry, to the C

terminus of vimentin (Fig. 1, A and B), as this is the preferred tagging site for avoiding aggregation and ensuring proper filament assembly (Herrmann et al., 1996; Usman et al., 2022). The FRB domain was fused to a kinesin motor to induce controlled movement. To trigger minus end–directed transport toward the microtubule-organizing center (MTOC) (Fig. 1 C), we employed the motor domain (amino acids 861–1321) of a moss kinesin-14–type VI kinesin-14b from the moss *Physcomitrella patens* (hereafter referred to as ppKin14) (Jonsson et al., 2015). The motor was tetramerized through a fusion with the leucine zipper domain of GCN4 (Nijenhuis et al., 2020) and tagged with a BFP or GFP fluorescent marker for visualization (Chen et al., 2022). This motor was selected because, by itself, it does not perturb endogenous transport in mammalian cells but can efficiently pull cargoes when artificially coupled to them (Nijenhuis et al., 2020). For plus end–directed transport toward the cell periphery (Fig. 1 C), we used a hemagglutinin (HA)-tagged dimeric motor domain fragment of the neuronal kinesin-1 KIF5A (amino acids 1–560) (Fig. 1, A and B). This motor was chosen for its demonstrated efficiency in pulling the highly interconnected and abundant ER network, which spans the entire cell (Özkan et al., 2021). Flexible glycine-serine linkers were inserted between the fusion protein domains to ensure proper protein folding and functionality.

To test these constructs, we transiently transfected them in COS-7 cells, fixed the cells after treatment with ethanol solvent with or without rapalog, and stained them for total vimentin, and in the case of the plus end–directed motor, also for the HA tag (Fig. 1, D and E). The mCherry-tagged vimentin strongly colocalized with total vimentin (Fig. 1, D and E; and Fig. S1 A), and the signal and the distribution of the latter appeared largely unperturbed in the absence of rapalog, though the fraction of the cell area occupied by vimentin was slightly larger in transfected cells than in untransfected control cells (Fig. 1 F). To assess the efficacy of motor recruitment, we conducted a 5-min treatment with and without rapalog, followed by super-resolution imaging of ppKin14 and KIF5A motors enhanced by antibody staining (Fig. S1, B and C). The results showed that, in the absence of rapalog, the motors did not associate with vimentin filaments: ppKin14 appeared diffuse, whereas KIF5A aligned with filaments that likely represented a stable microtubule subset, as published previously (Cai et al., 2009; Dunn et al., 2008; Reed et al., 2006). Upon rapalog addition, both motors densely populated vimentin filaments (Fig. S1, B and C). Treatment with rapalog for 1 h had no effect on vimentin distribution in untransfected cells; however, in transfected cells, the vimentin network collapsed completely (Fig. 1, D–F). In ppKin14-transfected cells, a single vimentin cluster was formed at the cell center, as expected. Surprisingly, the outcome in cells expressing KIF5A was quite similar, with a major vimentin cluster in the cell center and some additional, smaller clusters in the peripheral cytoplasm (Fig. 1, E and F). This conclusion was confirmed by analyzing the fraction of cell area occupied by vimentin, using cells with similar, moderate construct expression levels, which represented ∼50% of the total transfected cell population (Fig. 1, F–I). In the remaining 50% of the cells, the construct expression was either too high, causing vimentin pulling even without rapalog,

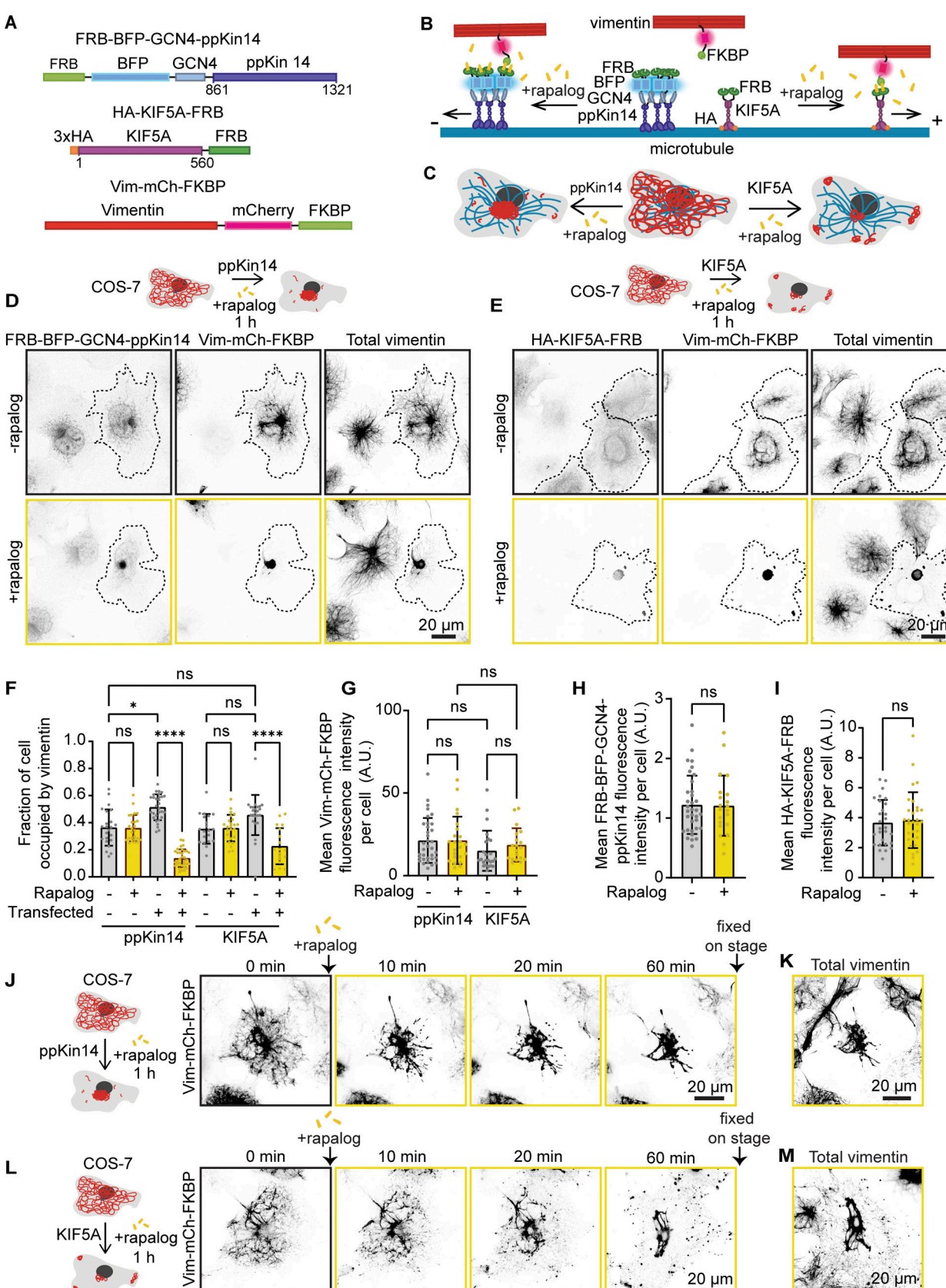

Figure 1. **Repositioning of the vimentin network by rapalog-induced recruitment of kinesin motors. (A)** A scheme of rapalog-induced hetero-dimerization constructs. A truncated motor domain of *P. patens* kinesin-14b (ppKin14, amino acids 861–1321) is fused to the FRB domain, along with a GCN4 leucine zipper for tetramerization and a BFP tag for detection. A fragment of human KIF5A (amino acids 1–560) is fused to an HA-tagged dimeric motor domain

and the FRB domain. The FKBP domain and mCherry were fused to the C terminus of vimentin. Flexible glycine-serine linkers separate protein domains. **(B and C)** Schemes illustrating vimentin repositioning triggered by rapalog-induced recruitment of kinesins at the level of a single microtubule (B) and in whole cells (C). Without rapalog, Vim-mCh-FKBP does not interact with either FRB-BFP-GCN4-ppKin14 or HA-KIF5A-FRB. Upon rapalog addition, FRB and FKBP heterodimerize, triggering motor attachment to vimentin filaments and their movement along microtubules. Minus-end-directed kinesins trigger vimentin clustering around the MTOC, whereas plus end–directed kinesins cause the formation of small peripheral clusters and a large perinuclear one. **(D and E)** Representative fluorescence images of COS-7 cells co-transfected with Vim-mCh-FKBP and FRB fusions of the indicated motors, with or without 1 h of rapalog treatment. The top panels show untreated cells, whereas the bottom panels display rapalog-treated cells. Transfected cells are outlined with dashed lines, and non-transfected cells serve as controls. Anti-vimentin and anti-HA antibodies detect total vimentin and HA-KIF5A-FRB, respectively. **(F)** Quantifications of the fraction of the cell area occupied by vimentin in untransfected and transfected cells with either minus- or plus-end motor constructs, with and without rapalog treatment. **(G–I)** Mean fluorescence intensities of Vim-mCh-FKBP intensity (G), FRB-BFP-GCN4-ppKin14 (H), and HA-KIF5A-FRB, detected with anti-HA antibody (I) per cell, normalized to untransfected cells. In (F–I), n = 19–34 cells per condition across three independent experiments. Plots indicate mean ± SD, with individual cell measurements shown as dots. ns, not significant; ∗, P < 0.05; ∗∗∗∗, P < 0.0001. Statistical significance was assessed using the Mann–Whitney t test for (H) and (I), while the Kruskal–Wallis test followed by Dunn's multiple comparisons test was applied for (F) and (G). **(J and L)** Live-cell imaging reveals the morphology of the vimentin network immediately before and at 10, 20, and 60 min after rapalog addition. Cells co-transfected with ppKin14 (J) or KIF5A constructs (L) show vimentin reorganization over time. **(K and M)** Cells shown in J and L, fixed and stained with antibodies against vimentin 60 min after rapalog treatment.

or insufficient to allow vimentin repositioning. Similar results were obtained in U2OS cells (Fig. S1, D and E), indicating that these effects were not cell line specific.

To observe vimentin relocalization in real time, we performed live-cell imaging before and after rapalog addition (Fig. 1, J and L). Vimentin repositioning toward microtubule minus ends with ppKin14 occurred rapidly, with strong vimentin clustering visible already within 10 min, likely due to the dense nature of the vimentin network in the cell center (Fig. 1 J and Video 1). The effect of KIF5A was slower (clustering visible within 20 min) and resulted in a pronounced tug-of-war (Videos 2 and 3), leading to the formation of vimentin clusters at microtubule crossroads at the cell periphery in addition to vimentin accumulation in the cell center (Fig. 1 L). After an hour of rapalog treatment, cells were fixed and stained for total vimentin, confirming its repositioning (Fig. 1, K and M). Taken together, our findings demonstrate that the rapalog-induced kinesin recruitment can trigger rapid vimentin displacement from most of the cytoplasm, with the minus end–directed ppKin14 organizing a single centrally positioned cluster and the plus end–directed KIF5A forming additional smaller clusters at the cell periphery.

**Light-induced vimentin repositioning**
We next modified the system by substituting the FKBP and FRB domains for the components of the improved Light-Induced Dimerizer (iLID) system (Guntas et al., 2015). This system consists of an optimized LOV2 domain derived from *Avena sativa* conjugated to the bacterial peptide SsrA (iLID) and the SsrA-interacting domain SspB. In the dark state, the SsrA in iLID is sterically blocked by LOV2, but upon LOV2 activation with blue light, SsrA becomes available for binding by SspB (Guntas et al., 2015). We fused the iLID to the two kinesin motors and linked the mCherry-tagged C terminus of vimentin to SspB$_{micro}$ (Fig. 2, A–C). The Vim-mCh-SspB successfully integrated into the endogenous vimentin network (Fig. 2 D). Upon a 5-min blue light illumination, U2OS cells transiently transfected with iLID-GFP-GCN4-ppKin14 and Vim-mCh-SspB exhibited motor recruitment to vimentin filaments (Fig. S2 A), similar to the FRB–FKBP system (Fig. S1 B). Longer blue light illumination was expected to induce vimentin relocalization, which was indeed observed. After 45 min of whole-cell blue light illumination, followed by

fixation and staining for total vimentin, relocalization was confirmed (Fig. 2, E and F). To assess the performance of the tool, we analyzed cells with comparable expression levels, evaluated by fluorescence intensity of the tags of the transfected constructs (Fig. 2, G and H). Among all transfected cells, ~40% were in the optimal range, ~40% were overexpressing the constructs too much, resulting in vimentin clustering without blue light illumination, and ~20% had too low expression, insufficient to trigger vimentin clustering in lit conditions. Within the optimized expression range, clustering of vimentin in the cell center was observed only in transfected cells and only upon blue light illumination (Fig. 2, E and F). This treatment had no effect on the cell area, indicating that vimentin clustering had no major effect on cell spreading (Fig. 2 I).

The advantage of light-induced heterodimerization is that it can also be used for subcellular manipulations. Indeed, by using local illumination with blue light, we could cause vimentin retraction in a part of the cell without affecting vimentin distribution in the rest of the cytoplasm (Fig. 2, J and K). Subsequent fixation and staining for vimentin confirmed that endogenous vimentin was also locally displaced along with the tagged vimentin (Fig. 2 L).

Another advantage of the iLID–SspB system compared with the rapalog-induced heterodimerization is that it is reversible. To evaluate reversibility, we first exposed transfected U2OS cells to 30 min of intermittent blue light, which resulted in vimentin clustering in the center of the cell, and then continued imaging the tagged vimentin without blue light to allow cells to recover (Fig. 2 M and Video 4). Cells required between 2.5 and 4 h to restore vimentin distribution (Fig. 2, M and N), likely through a combination of diffusion and active transport by the kinesin-1 motor, which transports vimentin filaments toward the cell periphery (Gyoeva and Gelfand, 1991; Robert et al., 2019). While some cells exhibited complete recovery, others regained only 40–70% of the initial vimentin density at the periphery (Fig. 2 N). This variability could not be explained by differences in expression levels (Fig. S2 B) but could be related to differences in the cell cycle phase or in the degree of vimentin compaction near the MTOC, potentially caused by cross-linking proteins trapped within the dense mesh (Petitjean et al., 2024; Wiche and Winter, 2011).

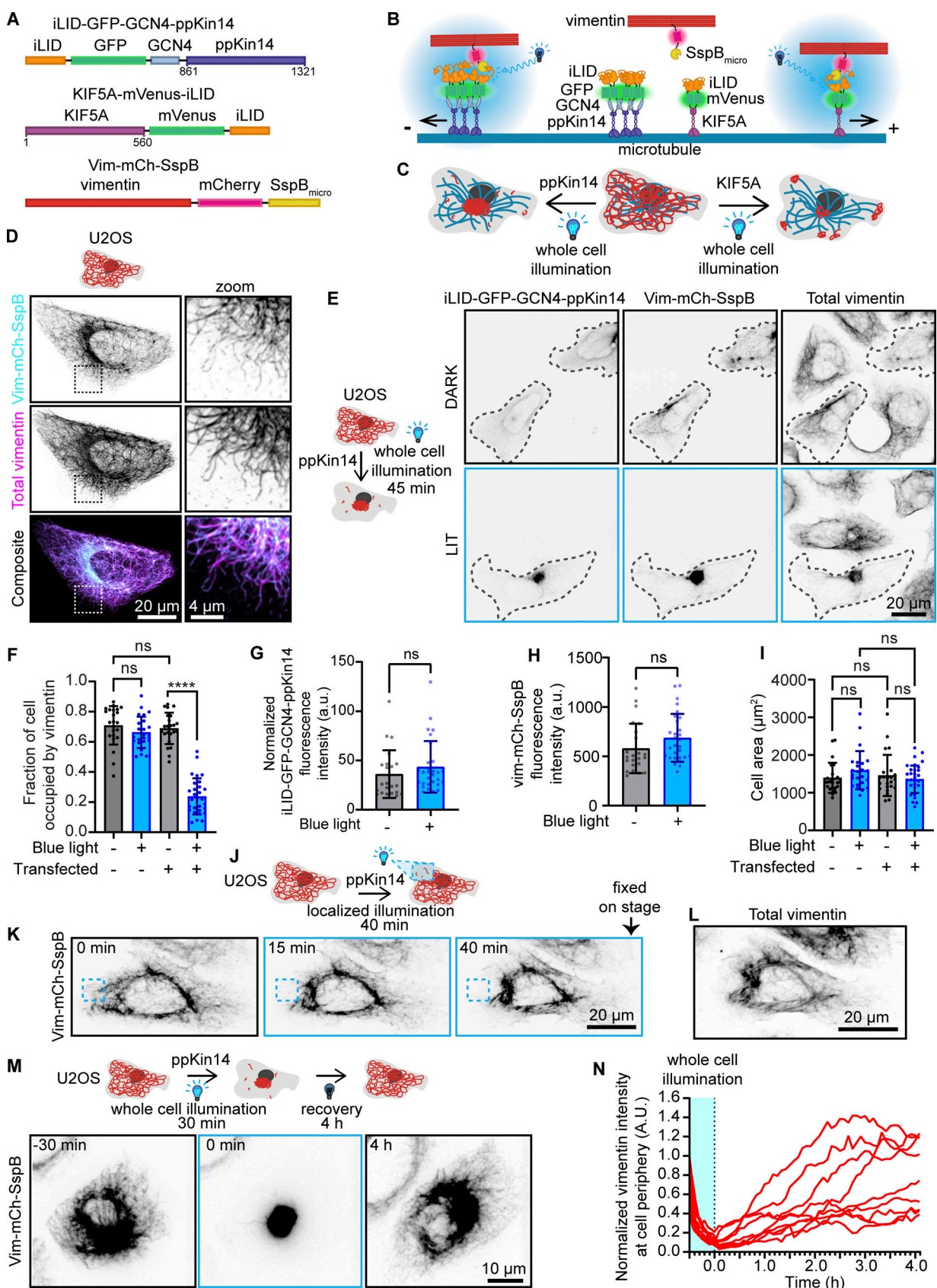

Figure 2. **Reversible vimentin repositioning by optogenetic recruitment of kinesin motors to vimentin. (A)** Schematic overview of optogenetic vimentin constructs. The constructs are similar to those shown in Fig. 1 A, except that FRB is substituted for iLID and FKBP for SSB_micro. GFP and mVenus are used as fluorescent markers in ppKin14 and KIF5A constructs, respectively. **(B and C)** Schematics illustrating the action of the optogenetic vimentin constructs.

**(B)** Upon blue light activation, the iLID module changes conformation, uncaging an SsrA peptide, which binds to SspB. By tagging vimentin with SspB and ppKin14 and KIF5A with iLID, vimentin can be inducibly pulled toward microtubule minus or plus ends, respectively. **(C)** Transfection of cells with Vim-mCh-SspB combined with one of the two kinesin constructs and blue light activation results in vimentin pulling to microtubule minus or plus ends, causing vimentin clustering either at the MTOC (ppKin14) or both the cell periphery and MTOC (KIF5A). **(D)** U2OS cells expressing the Vim-mCh-SspB construct. The over-expressed vimentin network is visualized by mCherry fluorescence, while total vimentin intensity is detected via anti-vimentin immunostaining. Images were captured using scanning confocal microscopy. **(E–I)** U2OS cells co-transfected with Vim-mCh-SspB and iLID-GFP-GCN4-ppKin14 were either fixed in a dark room (DARK) or exposed to 45 min of blue light (LIT) prior to fixation and staining for vimentin (total vimentin). **(E)** Representative images, with transfected cells outlined with a dashed line. **(F)** Quantification of the fraction of the cell area occupied by vimentin. **(G)** Mean iLID-GFP-GCN4-ppKin14 fluorescence intensity per cell, normalized to untransfected cells. **(H)** Mean cell Vim-mCh-SspB fluorescence intensity. **(I)** Mean total cell area. **(F–I)** $n$ = 23–29 cells per treatment analyzed over three experiments; bars show mean ± SD. ns, not significant; ****, P < 0.0001 by Mann–Whitney test (G and H) or Kruskal–Wallis with Dunn's test (F and I). **(J–L)** U2OS cells co-transfected with Vim-mCh-SspB and iLID-GFP-GCN4-ppKin14 were locally illuminated with 488-nm light pulses for 40 min before fixing and staining for vimentin. **(J)** Scheme of the experiment. **(K)** Stills with the region illuminated with blue light are indicated with a blue dashed box. **(L)** Vimentin staining in the cell shown in K was fixed after local illumination. **(M and N)** U2OS cells co-transfected with Vim-mCh-SspB and iLID-GFP-GCN4-ppKin14 were exposed to whole-cell 488-nm light pulses for 30 min to cluster vimentin, and then 488-nm pulses were stopped to allow cells to recover over 4 h. **(M)** Representative cell directly before light pulses (−30 min), after 30 min of 488-nm light pulsing (0 min), and after 4 h of recovery without blue light activation (4 h). **(N)** Quantification of Vim-mCh-SspB mean intensity at an ROI at the cell periphery over time; plot shows a line for every individual cell, $n$ = 9 cells analyzed over three independent experiments.

---

Finally, we also tested optogenetic vimentin repositioning in combination with the KIF5A-iLID fusion by whole-cell illumination and found that it worked similarly to KIF5A-FRB: vimentin formed some peripheral clusters, with a significant part of the filaments accumulating in the cell center (Fig. S2, C–E). Repositioning vimentin with a minus end–directed kinesin proved more effective than with a plus end–directed motor, likely due to the dense organization of the vimentin network in the perinuclear region.

### Vimentin clustering has little immediate effect on microtubules, actin, and focal adhesions

Considering that microtubules and vimentin can reciprocally affect each other (Gan et al., 2016; Prahlad et al., 1998; Schaedel et al., 2021), we next investigated whether vimentin clustering affects microtubule organization. To this end, COS-7 cells were co-transfected with Vim-mCh-FKBP and either FRB-BFP-GCN4-ppKin14 or HA-KIF5A-FRB, with non-transfected cells serving as controls (Fig. 3, A and B). After 1 h of treatment with or without rapalog, the cells were fixed and stained for tyrosinated tubulin, a marker of dynamic microtubules. The overall microtubule intensity did not differ between transfected and non-transfected cells, nor between rapalog-treated and -untreated conditions (Fig. 3 C). However, we noted that the microtubule network appeared slightly more disorganized in transfected cells, likely due to the overexpression of vimentin and motor proteins. This disorganization was not significantly different after pulling vimentin in either direction, as quantified by the ratio of non-radial to radial microtubules (Fig. 3 D).

Next, we examined whether stable microtubules would be affected by vimentin pulling. Given that COS-7 cells displayed variable levels of stable microtubules, as indicated by acetylated tubulin staining (data not shown), we performed these experiments in U2OS cells, which have much more consistent levels of stable microtubules. Cells were fixed after 1 h of treatment with or without rapalog. Cells co-expressing Vim-mCh-FKBP and FRB-BFP-GCN4-ppKin14 exhibited similar levels of acetylated tubulin compared with non-transfected cells, regardless of the treatment condition (Fig. 3, E and G). However, cells co-expressing Vim-mCh-FKBP with HA-KIF5A-FRB showed

significantly reduced acetylated tubulin levels even without rapalog addition (Fig. 3, F and G), and the expression of HA-KIF5A-FRB alone was sufficient to cause this effect (Fig. S3, A and B). This is consistent with a recent study showing that overexpression of kinesin-1 KIF5B fragment 1–560 reduces the number of stable, acetylated microtubules (Andreu-Carbó et al., 2024), which are the preferred tracks for this motor in cells (Cai et al., 2009; Dunn et al., 2008; Reed et al., 2006). Due to this strong effect of the KIF5A motor on stable microtubules, we decided to focus on minus end–directed pulling in subsequent experiments to avoid indirect effects of perturbing stable microtubules on the studied structures.

We next examined the effects of vimentin clustering on actin stress fibers and focal adhesions, structures that have been reported to be linked to vimentin IFs (Bhattacharya et al., 2009; Esue et al., 2006; Jiu et al., 2015; Mendez et al., 2010; Tsuruta and Jones, 2003; Venu et al., 2022, Preprint). 1 h after initiating rapalog-induced vimentin network relocalization by ppKin14, COS-7 cells were fixed and stained for total vimentin, F-actin, and the focal adhesion marker paxillin. Despite strong vimentin clustering, we observed no changes in the cell area, the appearance of F-actin structures, or the number of focal adhesions (Fig. 4, A–C). We repeated these experiments in U2OS cells, which have more prominent stress fibers and focal adhesions, but again observed no significant changes after vimentin repositioning (Fig. 4, D–F). This is consistent with the results of light-induced vimentin repositioning described above, where no effect on cell area was observed (Fig. 2 I). Our findings indicate that the clearance of vimentin from the cytoplasm and its clustering in the cell center do not have major short-term effects on microtubules, actin, or the ability of cells to adhere to substrate and spread.

### Co-dependency of keratin-8 and vimentin localization is cell type dependent

We next examined the impact of rapalog-inducible vimentin clustering on keratin IFs using COS-7, U2OS, and HeLa cells. All three cell lines showed endogenous expression of keratin-8, with the highest expression observed in HeLa cells (Fig. S4 A). In COS-7 cells, the keratin-8 and vimentin networks exhibited

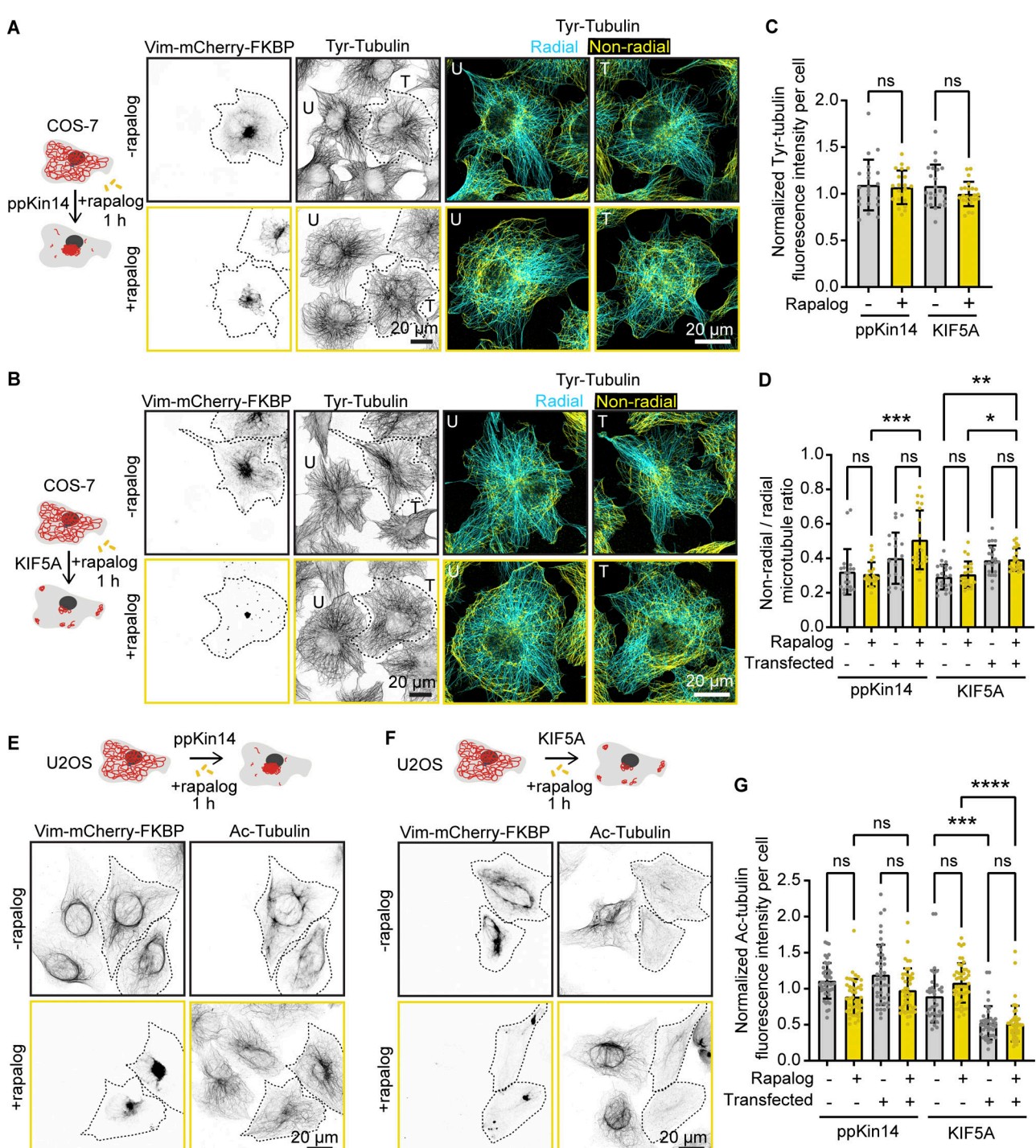

Figure 3.    **Effects of vimentin repositioning on the microtubule cytoskeleton. (A and B)** COS-7 cells were co-transfected with Vim-mCh-FKBP and either FRB-BFP-GCN4-ppKin14 (A) or HA-KIF5A-FRB (B), with non-transfected cells as controls. Transfected cells are outlined with a dashed line. After 1 h of treatment with or without rapalog, cells were fixed for analysis. Representative fluorescent images display the microtubule network, labeled with antibodies against tyrosinated α-tubulin (Tyr-tubulin). Microtubules are further color-coded to indicate radial (cyan) and non-radial (yellow) orientations. **(C)** Quantification of tyrosinated α-tubulin intensity in cells expressing Vim-mCh-FKBP along with either FRB-BFP-GCN4-ppKin14 or HA-KIF5A-FRB, normalized to non-transfected cells, with or without rapalog treatment. *n* = 21–23 cells analyzed across two independent experiments. **(D)** Quantification of the ratio of non-radial to radial tyrosinated microtubules in non-transfected cells and cells expressing Vim-mCh-FKBP with either FRB-BFP-GCN4-ppKin14 or HA-KIF5A-FRB, with or without rapalog treatment. *n* = 20–22 cells were analyzed across two independent experiments. **(E and F)** Representative images of U2OS cells stained for acetylated microtubules (Ac-tubulin), showing either untransfected cells or cells co-expressing Vim-mCh-FKBP with either FRB-BFP-GCN4-ppKin14 (E) or HA-KIF5A-FRB (F). Transfected cells are outlined with a dashed line. After 1 h of treatment with or without rapalog, cells were fixed for analysis. **(G)** Quantification of normalized acetylated microtubule intensity in untransfected U2OS cells and cells co-expressing Vim-mCh-FKBP with either FRB-BFP-GCN4-ppKin14 or HA-KIF5A-FRB, in the presence and absence of rapalog. *n* = 37–49 cells were analyzed across three independent experiments. Plots indicate mean ± SD, with individual cell measurements shown as dots. ns, not significant; *, P < 0.05; **, P < 0.01; ***, P < 0.001; ****, P < 0.0001 as assessed by Kruskal–Wallis test.

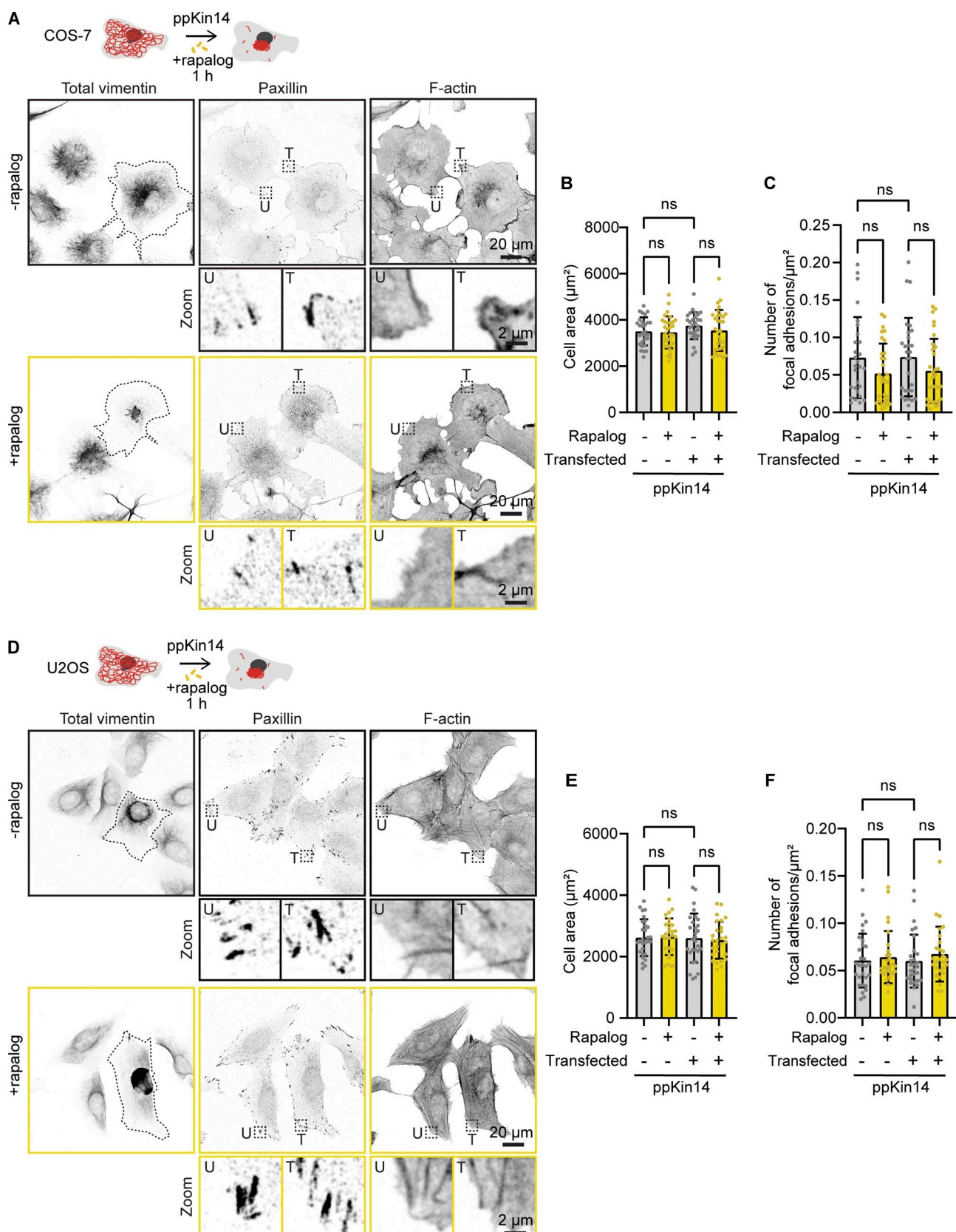

Figure 4. **Effects of vimentin repositioning on the actin cytoskeleton, cell spreading, and focal adhesions. (A–F)** COS-7 (A–C) or U2OS (D–F) cells were co-transfected with Vim-mCh-FKBP and FRB-BFP-GCN4-ppKin14, while non-transfected cells served as controls. Transfected cells are outlined with a dashed line. **(A and D)** After 1 h of treatment with or without rapalog, the cells were fixed and stained for total vimentin, paxillin, and actin using phalloidin. **(B and E)** Quantification of the total cell area based on the phalloidin staining in transfected (T) and untransfected (U) cells, with and without rapalog treatment. Dashed

boxes show regions enlarged in the zoom panels. **(C and F)** Quantification of the total focal adhesion number based on paxillin staining, normalized to cell area as determined by phalloidin staining in both transfected and untransfected cells, with or without rapalog treatment. Measurements were collected from $n$ = 28–31 cells in B and C and from $n$ = 27–29 cells in E and F across three independent experiments. The plots display the mean ± SD, with individual cell measurements represented as dots. ns, not significant, determined by Kruskal–Wallis analysis.

strong colocalization in the cell center, with keratin being more abundant than vimentin at the cell periphery (Fig. S4 B). Super-resolution imaging showed that the two proteins could be detected in the same filaments (Figs. 5, A and B; and S4, B and C). Overexpression of vimentin in COS-7 cells resulted in a more fragmented keratin-8 network and reduced its abundance at the cell periphery (Fig. 5 A). However, colocalization between keratin-8 and vimentin in transfected cells persisted, even following vimentin clustering (Fig. 5, A and B). Similarly, U2OS cells showed high vimentin-keratin-8 colocalization, with both proteins sometimes incorporated into the same filaments, although keratin-8 appeared more sparse (Fig. 5, C and D; and Fig. S4, D and E). Also in U2OS cells, vimentin pulling efficiently cleared keratin-8 from the cytoplasm (Fig. 5, C and D). These findings indicate a strong interdependence between the two IF types in COS-7 and U2OS cells.

In HeLa cells, both networks were dense around the nucleus, but the keratin-8 network also formed connections to the plasma membrane via desmosomes, as described previously (Fig. 5 E; and Fig. S4, F and G) (Jones and Goldman, 1985; Nishizawa et al., 2005; Schwarz et al., 2015). Unlike COS-7 and U2OS cells, vimentin and keratin-8 networks in HeLa cells appeared much more distinct and displayed a lower colocalization coefficient (Fig. 5 F), although some filaments containing both IF types were also observed (Fig. 5 E; and Fig. S4, F and G). Vimentin clustering did not affect the distribution of keratin-8 in HeLa cells, as reflected by a significant reduction in colocalization (Fig. 5, E and F). These results imply significant differences in the organization of vimentin and keratin-8 networks between cell types.

**Vimentin relocalization to the MTOC reduces cell stiffness**
Previous studies have demonstrated that cells lacking vimentin exhibit reduced stiffness and increased deformability (Mendez et al., 2014; Messica et al., 2017; Shaebani et al., 2022). Building on these findings, we aimed to investigate the effect of vimentin reorganization on cellular mechanical properties, with a particular focus on cell stiffness. To explore this, we used rapalog-induced vimentin reorganization in U2OS cells (Fig. 6 A) and measured cell stiffness using a spherical nanoindenter to determine the Young's modulus (Fig. S5). Indentations were performed at three locations within the perinuclear region (Fig. 6 B). Cells in which vimentin was pulled toward the cell center exhibited a reduction in Young's modulus of more than 60% compared with control cells (Fig. 6 C). This significant decrease in cell stiffness indicates that vimentin reorganization makes U2OS cells softer and more deformable, aligning with previous findings from vimentin knockout studies (Mendez et al., 2014; Messica et al., 2017; Shaebani et al., 2022).

**Vimentin pulling drives ER reorganization and reveals redundancy of ER-vimentin linkers**
Next, we investigated the effect of vimentin repositioning on intracellular membranes. Vimentin and the ER display a significant degree of colocalization, especially in the perinuclear area where ER sheets are present, as well as at the more peripherally located ER matrices (Cremer et al., 2023; Lynch et al., 2013; Nixon-Abell et al., 2016) (Fig. 7, A and B). At the cell periphery, ER intensity decreases, coinciding with the polygonal ER network and a sparser vimentin distribution (Fig. 7, A and B).

Using the optogenetic tool, we observed that ER sheets and matrices, but not tubules, moved along with vimentin, confirming their previously described direct connections (Cremer et al., 2023) (black arrows, Fig. 7 C and Video 5). Most of the vimentin and ER repositioning occurred within ~10 min (Fig. 7, C and D; and Video 5). While initially this resulted in a sparser tubular ER network at the cell periphery, over time, the network became denser, with smaller polygonal structures. This effect could also be observed in the ratio of perinuclear to peripheral intensity, where a subset of ER initially follows vimentin to the perinuclear region but then redistributes again toward the cell periphery (Fig. 7 D). It should be noted that while photobleaching of the ER channel was negligible, there was a 40% reduction in total Vim-mCh-SspB intensity over the course of the experiment (Fig. 7 E).

A similar pattern for ER reorganization was seen in fixed COS-7 cells using the rapalog-based tool to sequester vimentin (Fig. 7 F). After 1 h of rapalog-induced vimentin clustering, we observed two distinct phenotypes: in some cells, the ER remained sparse and mostly confined to the perinuclear area; in other cells, the ER reorganized, forming smaller polygons that extended toward the periphery (Fig. 7 F).

Next, we investigated the effect of vimentin pulling in the absence of RNF26, a recently described linker between ER sheets and vimentin (Cremer et al., 2023). We used U2OS WT and RNF26 knockout cells and performed localized light-induced vimentin relocalization inside a region of interest (ROI) at the cell periphery (blue dashed box, Fig. 8, A and B; and Videos 6 and 7). Surprisingly, even in the absence of RNF26, we observed during vimentin pulling a concomitant retraction of the ER sheets and matrices (Fig. 8 B and Video 7). This was also demonstrated by the reduction in fluorescence intensity of the ER marker inside the ROI accompanying the drop in vimentin intensity (Fig. 8 C). This implies the existence of additional linkers between the ER network and the vimentin cytoskeleton.

**Vimentin-mediated mitochondrial relocalization persists after pulling, whereas lysosomes rapidly recover**
We next investigated the effects of vimentin repositioning on other organelles. U2OS cells were used because in these

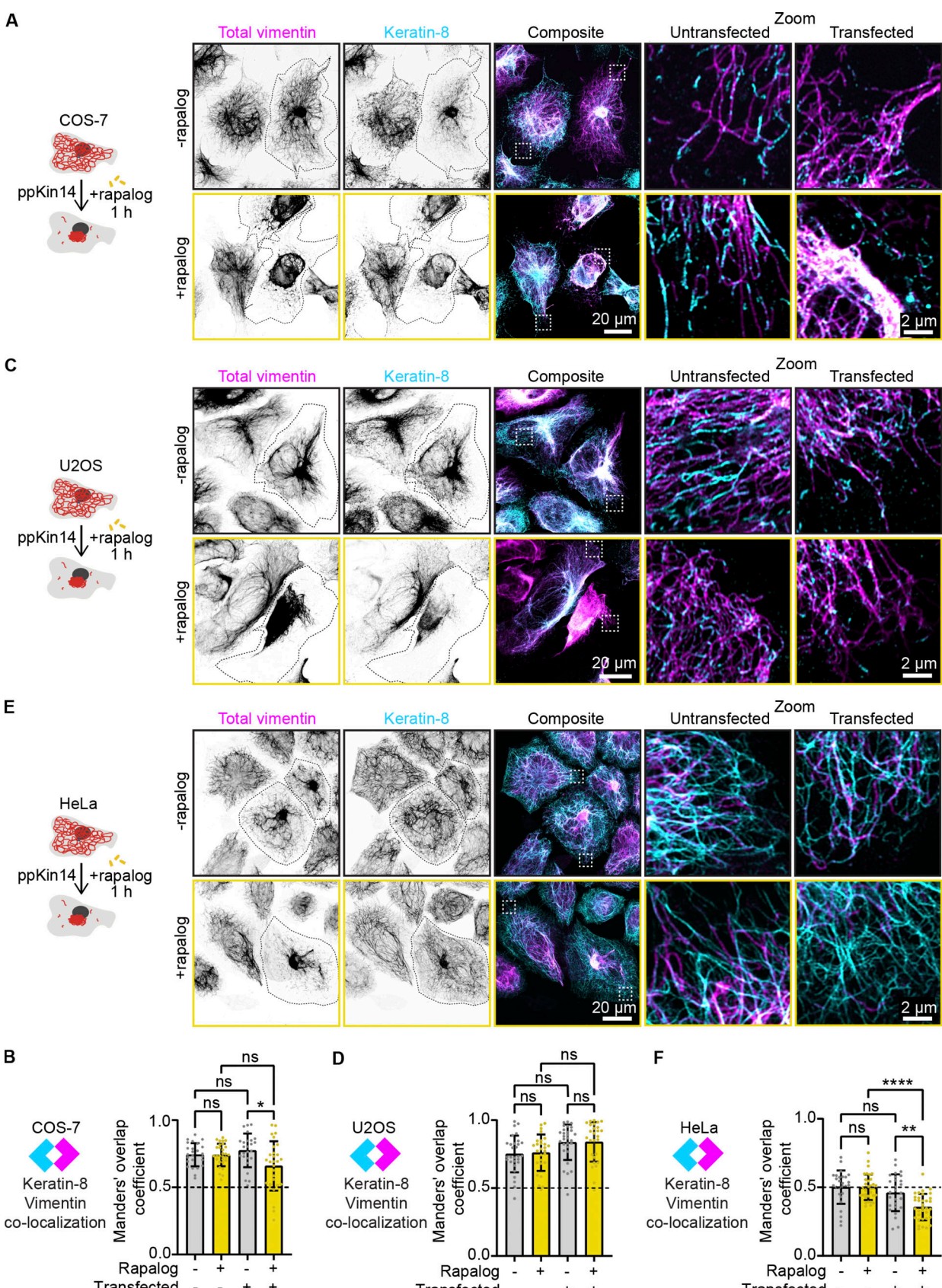

Figure 5. **Effects of vimentin clustering on the keratin-8 network across cell lines. (A, C, and E)** Indicated cell lines were co-transfected with Vim-mCh-FKBP and FRB-BFP-GCN4-ppKin14 constructs (transfected cells are outlined), while non-transfected cells served as controls. After 1 h of treatment with or without rapalog, the cells were fixed for analysis. Total vimentin and keratin-8 network intensities were measured after staining with anti-vimentin and

anti–keratin-8 antibodies, respectively. The white dashed box indicates the region of the cell enlarged in zoom images. Images were obtained with Airyscan microscopy. **(B, D, and F)** Colocalization analysis in COS-7 (B), U2OS (D), and HeLa (F) cells. The graphs represent Manders' coefficients of thresholded images, measured from 50.2 × 50.2-µm ROIs per COS-7 cell (B), 45.11 × 45.11-µm ROIs per U2OS cell (D), and 35.03 × 35.03-µm ROIs per HeLa cell (F). Graphs show mean ± SD, with individual cell measurements represented by dots. In B, data were collected from $n$ = 25–29 cells; in D, $n$ = 27–29 cells; and in F, $n$ = 27–34 cells across three independent experiments. ns, not significant; ∗, P < 0.05; ∗∗, P < 0.01; ∗∗∗∗, P < 0.0001 based on Kruskal–Wallis statistical analysis.

cells, organelles are distributed throughout the cell, whereas in COS-7 cells, most organelles are strongly clustered in the perinuclear region, and their relocalization with vimentin would be less obvious. Local vimentin pulling revealed that the clearance of vimentin led to a strong redistribution of mitochondria (Fig. 9 A and Video 8). Mitochondria remained co-clustered with vimentin also during longer (1 h) treatment periods, as observed in fixed cells after using the rapalog tool (Fig. 9 B). The fraction of mitochondria at the cell periphery (defined as the cell area outside the perinuclear region, which was empirically defined as a circle with a radius of 13.82 µm) was reduced in rapalog-treated transfected cells compared with untreated transfected cells (Fig. 9 C).

In contrast, lysosomes demonstrated a different behavior. In optogenetic experiments, lysosomes were initially pulled along with vimentin but quickly dissociated from the vimentin cluster and returned to the cell periphery (Fig. 9 D and Video 9). This was confirmed by the long-term (1 h) rapalog treatment and analysis of fixed cells, which showed no significant changes in lysosome distribution between treated and untreated cells (Fig. 9 E). The lysosome fraction at the cell periphery (outside the perinuclear region) remained the same, ~40% (Fig. 9 F). These data indicate that vimentin IFs are not a dominant factor in the steady-state distribution of lysosomes, and their transient displacement could be caused by their entrapment in the vimentin mesh or the presence of the membrane contact sites with other organelles, such as the ER or mitochondria.

## Discussion

In this study, we have introduced a rapid chemical and optogenetic approach to reposition the vimentin network, which can be precisely controlled in time and space. In cells with a radial microtubule organization and significant perinuclear clustering of vimentin, a minus end–directed kinesin could rapidly (within ~10 min) clear the cytoplasm of the majority of vimentin filaments without strongly perturbing either microtubules or the actin cytoskeleton. Also, a plus end–directed kinesin-1–derived tool disrupted the vimentin network, but it turned out to be less useful because it was slower and led to the formation of both peripheral and perinuclear vimentin clusters. This was likely because vimentin forms a very dense perinuclear cage that is difficult to pull apart; moreover, some of the preferred tracks of kinesin-1, stable microtubules, are positioned with their plus ends facing the cell periphery (Chen et al., 2022) and could target vimentin toward the nucleus.

The optogenetic approach for clustering vimentin was reversible, suggesting that pulling did not damage vimentin filaments. The recovery was, however, incomplete in some cells, possibly due to differences in the cell cycle stage or because the forces generated by kinesin-1 (Robert et al., 2019) may be insufficient to fully restore vimentin's original distribution. Furthermore, vimentin clustering might cause increased cross-linking via the tail domains (Aufderhorst-Roberts and Koenderink, 2019; Lin et al., 2010), or perhaps by proteins like plectin becoming trapped within the filament mesh (Foisner et al., 1988; Wiche and Baker, 1982; Wiche and Winter, 2011).

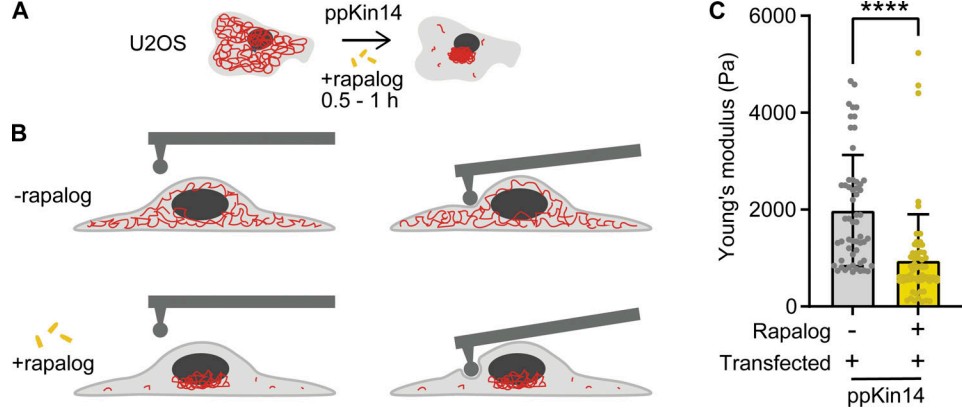

Figure 6.   **Effects of vimentin repositioning on cell stiffness. (A and B)** Schematic depiction of a control and a rapalog-treated U2OS cell co-transfected with Vim-mCh-FKBP and FRB-BFP-GCN4-ppKin14, before and during indentation in the perinuclear region using a spherical tip of 3.5-µm radius. Each cell was indented at three separate locations in the perinuclear region. **(C)** The graph presents the Young's modulus of cells co-transfected with Vim-mCh-FKBP and FRB-BFP-GCN4-ppKin14 constructs, with and without rapalog treatment. Data were collected from $n$ = 19–22 cells across two independent experiments. A total of 56 measurements were obtained from 19 cells without rapalog, and 63 measurements from 22 cells treated with rapalog. The plots display the mean ± SD, with each dot representing an individual cell measurement. ∗∗∗∗, P < 0.0001 via Mann–Whitney test.

Figure 7. **Effects of vimentin repositioning on ER morphology. (A and B)** Representative image of a COS-7 cell stained for endogenous calnexin (ER) and vimentin, and (B) intensity profile along the indicated line. Images were collected using STED microscopy. **(C–E)** U2OS cells co-transfected with Vim-mCh-SspB, iLID-GFP-GCN4-ppKin14, and Halo-KDEL were first imaged for 5 min without 488-nm pulsing (–5 min, 0 min), then imaged for 25 min with whole-cell 488-nm pulsing to induce optogenetic vimentin clustering (5, 10, 25 min). **(C)** Spinning disc confocal images from the experiment, with black dashed boxes

indicating the region of the cell enlarged in the zoom panels. **(D)** Plot showing ratio of perinuclear fluorescence intensity to intensity at the cell periphery. **(E)** Plot showing loss of total fluorescence signal over time. **(F)** Images of COS-7 cells expressing Vim-mCh-FKBP and FRB-BFP-GCN4-ppKin14. Transfected cells are outlined with dashed lines. After 1 h of treatment with or without rapalog, cells were fixed and stained for vimentin and calnexin. The images show the distribution of the ER network, visualized by calnexin staining, in both transfected (T) and untransfected (U) cells, in the presence and absence of rapalog. Dashed boxes show the regions that have been enlarged and shown as masks in the zoom images.

Our findings confirmed established functions of the vimentin network and provided new insights into its interactions with other cellular structures. Unlike previous approaches to perturb vimentin that often affected the organization of other cytoskeletal components such as microtubules, actin stress fibers, and focal adhesions (De Pascalis et al., 2018; Grin et al., 2012; Havel et al., 2015; Saldanha et al., 2023, *Preprint*; Swoger et al., 2022), our chemical and optogenetic manipulation of vimentin did not have a marked short-term effect on the appearance of these structures in the studied cell types in 2D cultures in glass. However, we have not investigated the effects of vimentin clustering on the dynamics of focal adhesions or the actin cytoskeleton. Rapidly migrating cells, such as fibroblasts, characterized by faster adhesion turnover rates (Gupton and Waterman-Storer, 2006; Mavrakis and Juanes, 2023) and larger focal adhesions (Kim and Wirtz, 2013), may react differently to vimentin perturbation. Given that our tools do not cause cell toxicity on a short time scale and can be applied locally, they could be used to study mechanistic details that were previously inaccessible, such as the local impact of vimentin on cell polarity and adhesion dynamics (Venu et al., 2022, *Preprint*). Furthermore, it will now be possible to distinguish direct effects of vimentin contacts with other cytoskeletal structures from the changes in signaling or gene expression induced by vimentin loss. Finally, since previous work has suggested that cells migrating on soft substrates are more sensitive to the loss of vimentin (Swoger et al., 2022), it would be interesting to test whether the same phenotypes can be reproduced with inducible vimentin clustering.

We identified cell type–specific differences in the interactions between vimentin and keratin. In COS-7 and U2OS cells, where the two IF types are co-expressed (termed hybrid cell states by Sha et al., 2019) but keratin levels are relatively low; super-resolution microscopy indicated that keratin and vimentin can co-assemble into the same filaments, and keratin-8 was pulled along with vimentin. In HeLa cells, where keratin levels are higher, co-assembly seemed to be less frequent, and vimentin clustering did not impact the keratin network. Colocalization of vimentin and keratin in certain epithelial cells has been detected previously (Robert et al., 2019; Velez-delValle et al., 2016) but has not received much attention, and it deserves further investigation.

Vimentin relocalization led to a strong reduction in stiffness of U2OS cells, consistent with findings from vimentin knockout cells (Mendez et al., 2014; Messica et al., 2017; Shaebani et al., 2022). Since we did not observe any changes in the actin or microtubule cytoskeleton in the time frame of the experiment but did find that keratin-8 co-clusters with vimentin in analyzed cells, we conclude that IFs by themselves make a major contribution to cell rigidity.

We also examined the relationship between vimentin and the ER, building on previous findings (Cremer et al., 2023; Lynch et al., 2013). Colocalization of the vimentin and ER networks, particularly ER matrices (Nixon-Abell et al., 2016) and sheets during acute vimentin displacement, confirmed their physical interaction. Notably, even in the absence of the vimentin-ER linker, RNF26, ER displacement together with vimentin was still observed, implying the existence of other linkers. Potential candidates for this function are nesprins, such as nesprin-3, which can connect to vimentin through plectin (Ketema et al., 2013; Nery et al., 2008). While these proteins are mostly concentrated in the nuclear envelope by binding to the SUN domain proteins at the inner nuclear membrane (Kim et al., 2015), they are transmembrane proteins that could be present in other ER compartments.

In addition to the ER, other organelles were affected by vimentin pulling. Mitochondria moved together and remained co-clustered with vimentin, consistent with previous observations that vimentin influences mitochondrial positioning (Nekrasova et al., 2011). Meanwhile, lysosomes showed only transient interactions with vimentin, as a fraction of lysosomes co-clustered with vimentin during pulling but rapidly regained their original distribution. This suggests a much weaker interaction, probably mediated by the contacts with the ER network (Kilpatrick et al., 2013; Özkan et al., 2021) or by entrapment in the vimentin mesh.

In summary, our vimentin-pulling tool is versatile and can be applied across various cell types. Future optimizations, such as tagging and pulling endogenous vimentin using a knock-in approach or generation of clonal stable cell lines, could help minimize side effects associated with vimentin and motor protein overexpression and improve the reproducibility of reversible pulling assays. This tool can shed light on cytoskeletal dependencies, cell protrusion and migration, and distinguish the immediate effects of vimentin reorganization on cell architecture from the long-term impact on signaling pathways and gene expression.

## Materials and methods

Chemicals used in this paper are given in Table 1, cell lines in Table 2, plasmids in Table 3 and software and plugins in Table 4.

### Cell culture and transfections

COS-7, HeLa, U2OS WT, and RNF26 knockout cells were cultured in DMEM (Sigma-Aldrich), supplemented with 10% FCS, 100 U/ml penicillin, and 100 µg/ml streptomycin (1% Pen Strep; Sigma-Aldrich). Cells were maintained at 37°C in a humidified atmosphere with 5% $CO_2$ and were regularly tested for mycoplasma contamination. For immunostaining, cells were plated on 18-mm coverslips, while 25-mm coverslips or gridded dishes

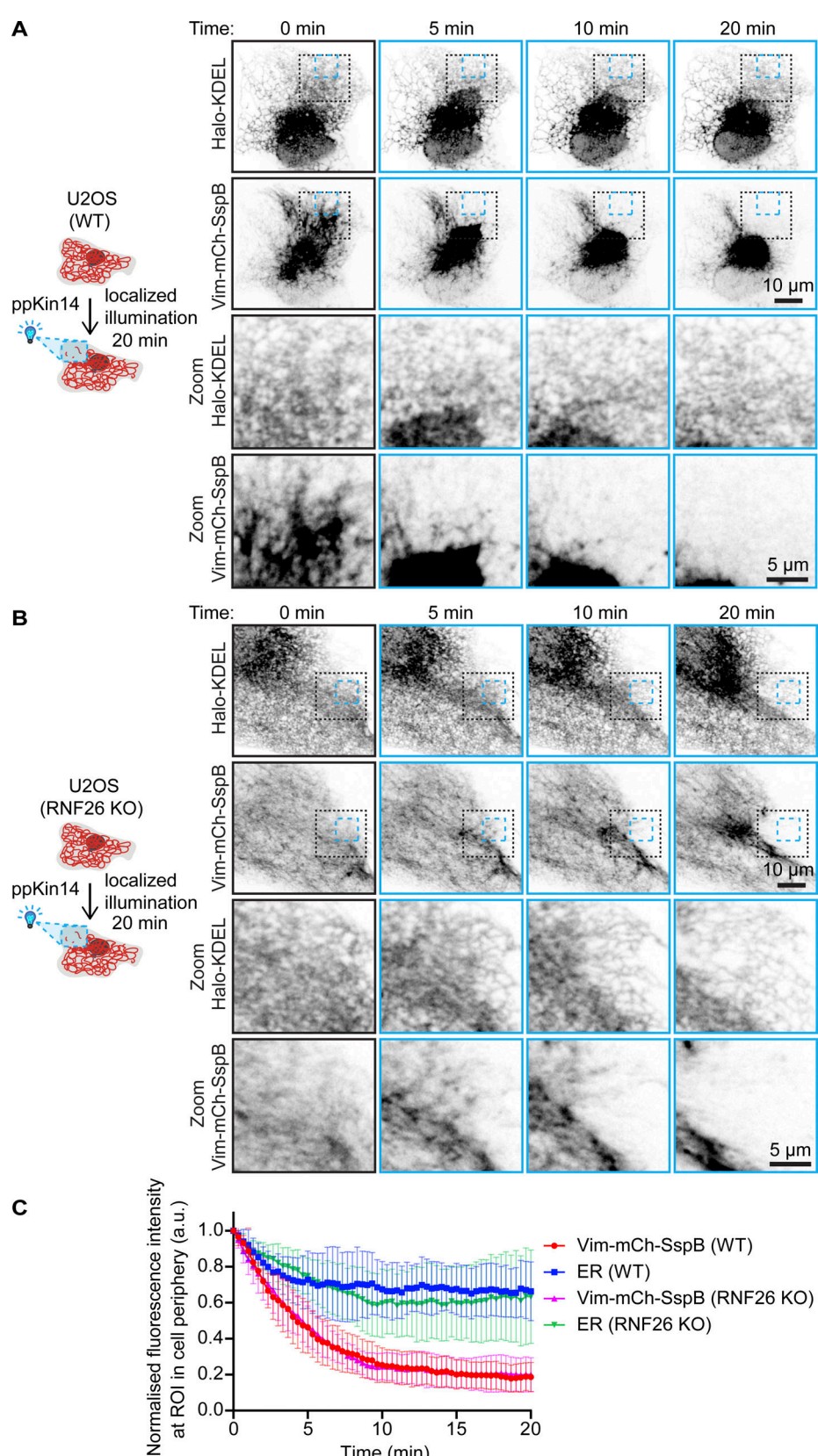

Figure 8.   **Localized optogenetic vimentin repositioning displaces dense ER matrices independent of RNF26. (A–C)** U2OS WT (A) and RNF26 knockout (KO) (B) cells co-transfected with Vim-mCh-SspB, iLID-GFP-GCN4-ppKin14, and Halo-KDEL were pulsed with 488-nm light inside an ROI at the cell periphery (blue dashed box) for 20 min to locally displace vimentin. **(A and B)** Example images shown before blue light pulsing (0 min) and 5, 10, and 20 min after localized blue light application. The lack dashed box indicates the region of the cell enlarged in the zoom images. **(C)** Mean fluorescence intensity of Vim-mCh-SspB and Halo-KDEL (ER) inside the ROI pulsed with blue light was quantified over time and normalized to the first time point after subtracting background. Graph shows mean ± SD, $n$ = 8–9 cells analyzed per group over three independent experiments.

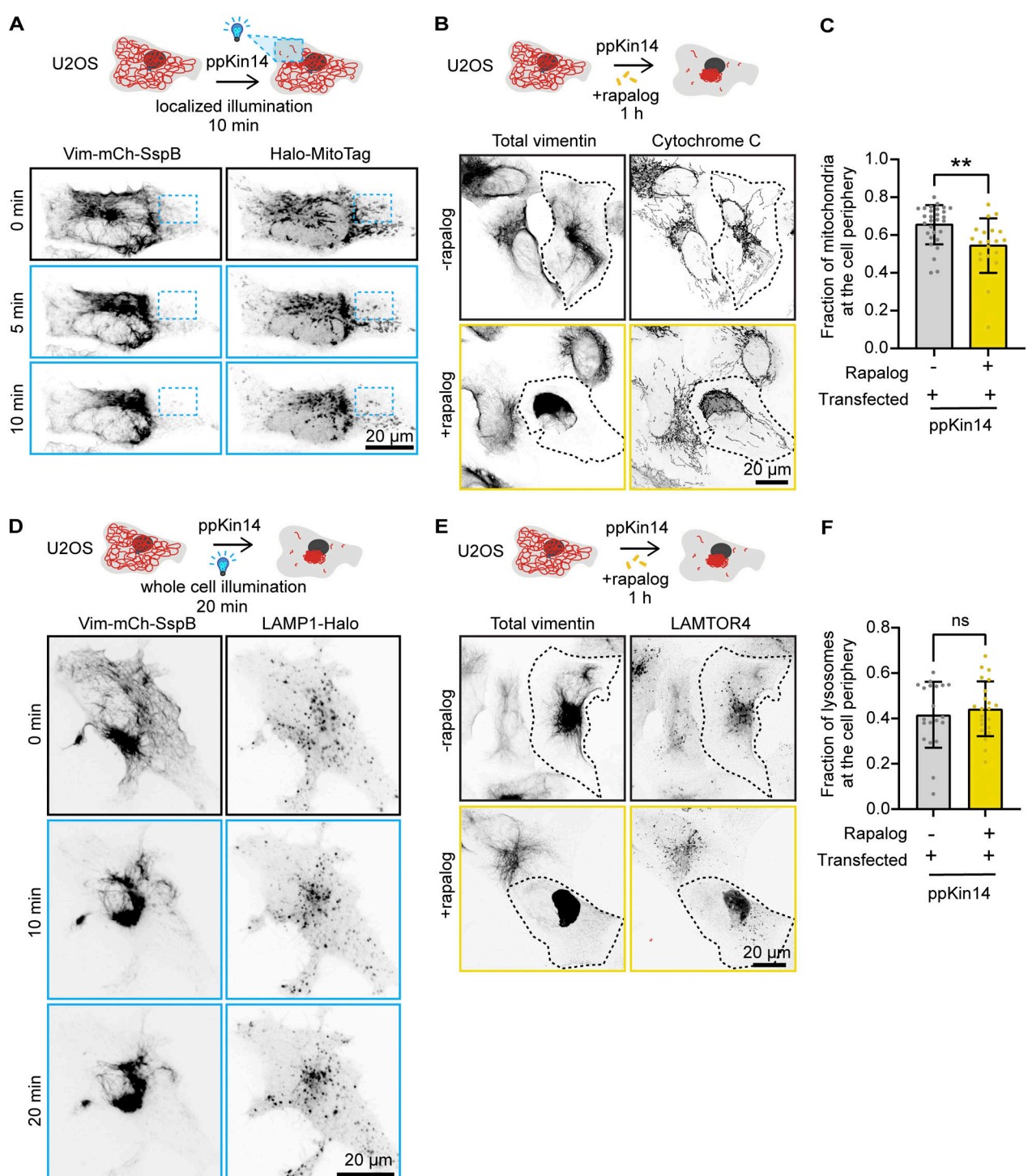

Figure 9. **Effects of vimentin clustering on the distribution of mitochondria and lysosomes. (A and D)** U2OS cell, co-transfected with Vim-mCh-SspB, iLID-GFP-GCN4-ppKin14, and Halo-MitoTag to label mitochondria (A) or LAMP1-Halo to label lysosomes (D), locally illuminated with 488-nm pulses inside an ROI (blue dashed box) for 10 min (A) or with global illumination for 20 min (D). The cell is shown before application of blue light (0 min) and at the indicated time points after blue light pulsing. **(B and E)** Imaging of U2OS cells transfected with Vim-mCh-FKBP and FRB-BFP-GCN4-ppKin14 constructs. After 1 h of treatment with or without rapalog, cells were subjected to immunostaining for total vimentin and cytochrome C to label mitochondria (B) or LAMTOR4 to label lysosomes (E). Transfected cells are indicated by a dotted outline. Images were acquired with Airyscan mode. **(C and F)** The distribution of mitochondria (C) or lysosomes (F) relative to the MTOC. Organelles were detected using the ComDet plugin, and their distance from the MTOC was calculated using a radius plugin. The graph shows the percentage of mitochondria or lysosomes located in the peripheral region, which was defined as an area beyond a 13.81-μm radius from the MTOC, with each dot representing data from an individual cell. Measurements were collected from n = 20–29 cells in C and from n = 20–22 cells in F across three independent experiments. The plots display the mean ± SD, with individual cell measurements represented as dots. ns, not significant; **, P < 0.01 by Mann–Whitney test.

Table 1. **Reagents**

| Chemicals | Company | Catalog number |
|---|---|---|
| SiR-tubulin | Spirochrome | SC014 |
| Janelia Fluor 646 HaloTag ligand | Promega | GA1120 |
| FuGENE 6 | Promega | E2692 |
| Pen/strep | Gibco | 15140-122 |
| Gibson assembly master mix | Thermo Fisher Scientific | A46629 |
| ProLong glass antifade mountant | Thermo Fisher Scientific/Life Tech | P36984 |
| 16% PFA methanol-free | Fisher Emergo | 28908 |
| Aqueous glutaraldehyde EM grade, 10% | Electron Microscopy Sciences | 16110 |
| A/C heterodimerizer (rapalog) | Takara | AP21967 |
| A/C heterodimerizer (rapalog) | Bio Connect | 635056 |

(Lonza) were used for live-cell imaging. Gridded dishes were used if cells needed to be first imaged live and then found back after fixing and staining. Transient transfections were performed using the Fugene 6 transfection reagent, following the manufacturer's protocol. For optogenetic construct transfections in 6-well plates, 2 µg Vim-mCh-SspB and 500 ng iLID-GFP-GCN4-ppKin14 or 200 ng KIF5A-mVenus-iLID was used per well; for 12-well plates, the quantity was halved. For chemical pulling experiments in 6-well plates, each well was transfected with 500 ng of Vim-mCh-FKBP and either 250 ng of FRB-BFP-GCN4-ppKin14 or HA-KIF5A-FRB. In 24-well plates, the quantities were also halved. Cells were subjected to rapalog treatments, live-cell imaging, or fixation 1 day after transfection.

### Drug treatment

For rapalog-inducible dimerization experiments, cells were plated on 12-mm cover glasses and co-transfected the following day with Vim-mCh-FKBP and either FRB-BFP-GCN4-ppKin14/FRB-GFP-GCN4-ppKin14 or HA-KIF5A-FRB constructs. 24 h after transfection, cells were treated with 400 nM rapalog (AP21967 A/C heterodimerizer; Takara) for 5 min or 1 h.

### Plasmids and cloning

The Vim-mCh-FKBP construct (#240423; Addgene) was generated using PCR-based cloning combined with Gibson assembly. The mCherry-N1 construct served as the backbone, with vimentin amplified from human cDNA encoding isoform 1 of vimentin (corresponding to CCDS7120, 466 amino acids) and the FKBP domain amplified from the FKBP-mCherry-CAMSAP2 plasmid (Chen et al., 2022). Glycine-serine linkers were included between protein segments. The FRB-BFP-GCN4-ppKin14 (#240426; Addgene) and FRB-GFP-GCN4-ppKin14 (#240427; Addgene) were described previously (Chen et al., 2022). To make Vim-mCh-SspB (#240421; Addgene), vimentin was first amplified from human cDNA encoding isoform 1 (corresponding to CCDS7120, 466 amino acids) and cloned into the KIF1A(1–365)-GFP-SspB plasmid (#174629; Addgene) (Nijenhuis et al., 2020), replacing KIF1A, cut out using AscI and XbaI restriction sites, by Gibson assembly; next the GFP was replaced by the mCherry

from SspB-mCh-p60 (#190168; Addgene) (Meiring et al., 2022) by restriction enzyme cloning using XbaI and NheI restriction sites. The iLID-GFP-GCN4-ppKin14 (#240420; Addgene) was generated from SspB-GFP-GCN4-ppKin14 (#174640; Addgene) (Nijenhuis et al., 2020) by restriction enzyme cloning using AscI and XbaI restriction sites to replace the SspB with the iLID from iLID-mCl3-Tau0N4R (Meiring et al., 2022). The KIF5A-mVenus-iLID (#240422; Addgene) was cloned by first substituting the SspB in KIF1A(1-365)-Venus-SspB (#174635; Addgene) (Nijenhuis et al., 2020) for an iLID from EB3N-VVDfast-mCl3-iLID (#190165; Addgene) (Meiring et al., 2022) by restriction enzyme cloning using NheI and NdeI restriction sides and subsequently replacing the KIF1A with KIF5A amplified from 3xHA-KIF5A-FRB (#240425; Addgene) using AscI and XbaI restriction sites and Gibson assembly. The Halo-MitoTag (#240424;Addgene) was made by cutting out the Rab6a from Halo-Rab6a (#190171; Addgene) (Meiring et al., 2022) using KpnI and BamHI and inserting the C-terminal tail of the actA protein as a mitochondrial-targeting sequence (Zhu et al., 1996) 5'-TTA ATTCTTGCAATGTTAGCTATTGGCGTGTTCTCTTTAGGGGCG TTTATCAAAATTATTCAATTAAGAAAAAATAAT-3' by Gibson assembly. Halo-KDEL and LAMP1-Halo were previously described by Meiring et al. (2022). 3xHA-KIF5A-FRB (#240425; Addgene) was kindly provided by Ginny Farías (Utrecht University, Utrecht, The Netherlands). mCherry-N1, KIF1A-GFP-SspB, KIF1A(1–365)-mVenus-SspB, (1–365)-GFP-SspB, and SspB-GFP-GCN4-ppKin14 were gifts from Lukas Kapitein (Utrecht University, Utrecht, The Netherlands). All constructs were confirmed by DNA sequencing.

Table 2. **Cell lines**

| Cell lines | Source | Identifier |
|---|---|---|
| HeLa Kyoto | Narumiya S., Kyoto University | CVCL_1922 |
| COS-7 | ATCC | CVCL_0224 |
| U2OS | ATCC | CVCL_0042 |
| U2OS RNF26 KO | Berlin I., Leiden University Medical Center Cremer et al. (2023) | |

**Table 3. DNA Plasmids**

| Recombinant DNA | Reference | Addgene ID |
|---|---|---|
| FRB-BFP-GCN4-ppKin14 | Chen et al. (2022) | #240426 |
| FRB-GFP-GCN4-ppKin14 | Chen et al. (2022) | #240427 |
| HA-KIF5A-FRB | This paper | #240425 |
| Vim-mCh-FKBP | This paper | #240423 |
| iLID-GFP-GCN4-ppKin14 | This paper | #240420 |
| KIF5A-mVenus-iLID | This paper | #240422 |
| Vim-mCh-SspB | This paper | #240421 |
| Halo-KDEL | Meiring et al. (2022) | |
| Halo-mitoTag | This paper | #240424 |
| LAMP1-Halo | Meiring et al. (2022) | |

**Table 4. Software and plugins**

| Software/Plugin | Source | Identifier |
|---|---|---|
| FIJI | ImageJ | https://imagej.net/software/fiji/downloads |
| ComDet plugin | Fiji | https://github.com/ekatrukha/ComDet |
| Radiality map plugin | Fiji | https://github.com/UU-cellbiology/radialitymap |
| Radius plugin | Fiji | https://gist.github.com/ekatrukha/105553627f1bee01367faae153bfe5c0 |
| MtrackJ | Smal et al. (2008) | N/A |
| Sigma-Aldrich plot | Systat Software, Inc. | 7 |
| Excel | Microsoft | Office16 |
| Prism | GraphPad | 10.1.1(323) |
| MetaMorph | Molecular devices | 7.10.2.240 |
| Dataviewer | Optics11 | 2 |

## Antibodies, immunofluorescence staining, and western blotting

Different fixation methods were employed depending on the target protein for immunofluorescence experiments. For vimentin, microtubules, and keratin-8, cells were fixed on ice in ice-cold methanol for 5–15 min. For calnexin staining, cells were fixed with 0.1% glutaraldehyde, 4% PFA, and 4% sucrose prewarmed at 37°C for 10 min at room temperature. Cytochrome C, Lamtor4, phalloidin, and paxillin staining were performed on cells fixed in 4% PFA in PBS for 15 min at room temperature. After fixation, cells were washed thrice with PBS and, unless methanol fixed, permeabilized with 0.2% Triton X-100 in PBS for 2.5 min. Following another washing step with PBS, cells fixed with glutaraldehyde were quenched with 100 mM sodium borohydride in PBS three times, 5 min each. After another PBS washing step, cells were blocked in 2% BSA in PBS for an hour at room temperature. The incubation of the primary and secondary antibodies was done at room temperature for an hour, with the antibodies diluted in 2% BSA in PBS. For F-actin labelling, Phalloidin conjugated to Alexa Fluor 647 (#A22287; Thermo Fisher Scientific) was added with the secondary antibodies. Between the primary and secondary antibody incubations, there was a washing step with 0.05% Tween-20 in PBS. After the last wash, coverslips were mounted with Pro-Long Gold mounting media.

The primary and secondary antibodies used in this study were as follows is given in Tables 5 and 6.

For western blotting, cells were harvested at 90–100% confluency from a 10-cm dish. Lysis was performed using RIPA buffer, supplemented with protease and phosphatase inhibitors (Roche). Proteins were separated on 10% polyacrylamide gels and transferred onto 0.45-µm nitrocellulose membranes (Sigma-Aldrich). Membrane blocking was performed in 2% BSA in PBS for 30 min at room temperature. The membrane was initially exposed to primary antibodies overnight at 4°C, then washed three times with 0.05% Tween-20 in PBS. Next, IRDye 680LT anti-mouse (#926-68020; LI-COR Biosciences) and IRDye 800CW anti-rat (#926-32219; LI-COR Biosciences) secondary antibodies, each diluted 1:1,000 in 2% BSA in PBS, were added to the membrane. The incubation was carried out for 1 h at room temperature, followed by washing with 0.05% Tween-20 in

PBS. Finally, membranes were imaged on an Odyssey CLx infrared imaging system (Image Studio version 5.2.5, LI-COR Biosciences).

## HaloTag labelling

HaloTag dye JF646 (Promega) was diluted in DMSO to 200 µM, aliquoted, and stored at −20°C. Aliquoted HaloTag dye was diluted 1:10,000 in pre-warmed cell media, vortexed for 5 s, and incubated with cells overnight.

## Microscopy

### Leica SP8 confocal microscopy

We conducted confocal or gated STED imaging using a Leica TCS SP8 STED 3X microscope, controlled by LAS X software. The setup included an HC PL APO 100x/1.4 oil STED WHITE objective, with white laser excitation at 488, 577, and 633 nm and depletion at 775 nm using a pulsed laser. Images were captured in 2D STED mode with a vortex phase mask. The Leica PMT and HyD hybrid detector were used, with a time gate range of 0.5–9 ns, and depletion laser power was set at 50%. Images were primarily acquired using confocal microscopy; instances where STED was used are explicitly noted in the figure legends.

### Airyscan confocal microscopy

A Carl Zeiss LSM880 Fast AiryScan microscope equipped with 405-nm, Argon multiline, 561-nm, and 633-nm lasers as well as AiryScan and PMT detectors was utilized for fixed confocal imaging. Samples were imaged using an Alpha Plan-APO 100x/1.46 Oil DIC VIS objective, and the microscope was operated by ZEN 2.3 software. Images acquired using Airyscan mode are indicated in the figure legends.

### Spinning disc microscopy

Spinning disc microscopy was utilized for all live-cell imaging experiments. This was performed on a Nikon Eclipse Ti2-E

Table 5.  **Primary antibodies**

| Antibody | Host species | Company | Catalog number | Dilution IF/WB |
| --- | --- | --- | --- | --- |
| Actin c4 | Mouse | Sigma-Aldrich/Merck | MAB1501 | -/1:1,000 |
| Calnexin | Rabbit | Abcam | 22595 | 1:250 |
| Cytochrome C | Mouse | BD Pharmingen | 556432 | 1:200 |
| Cytokeratin-8 (K8/KRT8) | Rat | DSHB, TROMA-I | AB_531826 | 1:12/1:100 |
| GFP | Mouse | Sigma-Aldrich | 11814460001 | 1:200 |
| HA | Rat | Roche | 11867423001 | 1:200 |
| HA | Mouse | Covance | MMS-101R | 1:200 |
| LAMTOR4 (D6A4V) | Rabbit | Cell Signaling | 12284 | 1:200 |
| Paxillin | Mouse | BD Biosciences | 610619 | 1:200 |
| Tubulin-acetylated | Mouse | Sigma-Aldrich | T7451-200UL | 1:200 |
| Tubulin-acetylated (Lys40) | Rabbit | Bioke | 5335S | 1:500 |
| Tubulin-alpha YL1/2 (tyrosinated) | Rat | Pierce/Thermo Fisher Scientific | MA1–80017 | 1:400 |
| Vimentin | Chicken | Abcam | ab24525 | 1:200 |
| Vimentin | Rabbit | Abcam | ab92547 | 1:400 |
| Vimentin V9 | Mouse | Sigma-Aldrich | V6630 | 1:200/1:1,000 |

inverted research microscope with a Nikon Perfect Focus System, a Nikon Plan Apo VC 100x N.A. 1.40 oil objective, and a spinning disk confocal scanner unit (CSU-X1-A1; Yokogawa). The system featured a Photometrics PRIME BSI back-illuminated sCMOS camera, an ASI motorized stage with piezo plate MS-2000-XYZ, and MetaMorph 7.10 software. It was equipped with three lasers: a 488 nm, 150 mW Vortran Stradus 488; a 561 nm, 10 mW Coherent OBIS 561-100LS; and a 639 nm, 150 mW Vortran Stradus 639. For imaging, we used the ET-GFP filter set (49002; Chroma) for GFP-tagged proteins, the ET-mCherry filter set (49008; Chroma) for mCherry-tagged proteins, and the ET-Cy5 filter set (49006; Chroma) for HaloTag JF646-tagged proteins. To maintain the cells at 37°C, we used a stage incubator (STXG-PLAMX-SETZ21L; Tokai Hit). For localized illumination experiments, the iLas FRAP system controlled with iLas software (Gataca Systems) was used.

### Optogenetic vimentin pulling
For whole-cell illumination-fixed experiments, cells were either protected from light and fixed in a dark room with only red and green light (DARK) OR placed on a blue LED array and illuminated for 45 min (800 µW/cm$^2$) (LIT). Live-cell imaging experiments were performed on a spinning disc microscope, using a 488-nm laser to activate iLID-SspB. For whole-cell activation, either one 500-ms exposure (0.25 mW, 300 µW/cm$^2$) every 4 s or one 1-s exposure (0.25 mW, 300 µW/cm$^2$) every 30 s was used. For localized activation, a FRAP unit was used to apply one localized scan with a duration of 113 ms (3 µW, 700 µW/cm$^2$) every 4 s.

### Cell stiffness measurements
U2OS cells were seeded in 35-mm glass-bottom dishes (World Precision Instruments) and allowed to adhere overnight. Cells were subsequently transfected as described above, and stiffness measurements were performed the following day. The Young's modulus of cells before and after vimentin pulling was measured using a Chiaro Nanoindenter (Optics11 Life). We indented cells with a spherical tip with a 3.5-µm radius, attached to a flexible cantilever with a stiffness of 0.027 N/m. Indentations were

Table 6.  **Secondary antibodies**

| Antibody | Host species | Company | Catalog number | Dilution |
| --- | --- | --- | --- | --- |
| Anti-mouse Alexa Fluor - 594 | Goat | Jackson ImmunoResearch | 115-585-166 | 1:200 |
| Anti-chicken Alexa Fluor - 594 | Goat | Thermo Fisher Scientific | A11042 | 1:200 |
| Anti-mouse Alexa Fluor - 488 | Goat | Thermo Fisher Scientific | A11029 | 1:200 |
| Anti-mouse DyLight - 405 | Goat | Jackson ImmunoResearch | 115-475-166 | 1:200 |
| Anti-rabbit Alexa Fluor - 488 | Goat | Thermo Fisher Scientific | A32731 | 1:200 |
| Anti-rabbit Alexa Fluor - 594 | Goat | Thermo Fisher Scientific | A11012 | 1:200 |
| Anti-rat Alexa Fluor - 488 | Donkey | Thermo Fisher Scientific | A21208 | 1:200 |
| Anti-rat Alexa Fluor - 647 | Goat | Thermo Fisher Scientific | A21247 | 1:200 |

made with a loading rate of 2 μm/s to a depth of 1.5 μm. For vimentin-pulled cells, we waited 30 min after adding rapalog before performing indentation measurements. The Chiaro Nanoindenter was mounted on a THUNDER epifluorescence microscope (Leica), allowing us to identify transfected cells. Cells selected for indentation were successfully co-transfected with Vim-mCh-FKBP and FRB-BFP-GCN4-ppKin14 Young's modulus was calculated from measured force–distance curves using the Hertzian contact model by the Optics11 Life Dataviewer software (version 2). Each cell was indented at three separate locations in the perinuclear region. Measurements without a distinct contact point or with an otherwise unreliable model fit (<0.9 $R^2$) were regarded as outliers and excluded from the analysis.

## Quantification and statistical analysis

FIJI software was utilized to adjust contrast levels and perform background corrections for image and video preparation. Fluorescence intensity quantification was also carried out in FIJI, with subsequent data processing and normalization completed in Excel. Statistical analysis and graph generation were performed using GraphPad Prism 10.1.1 (323). The Mann–Whitney test was applied for comparisons between two conditions, whereas the Kruskal–Wallis test, followed by Dunn's multiple comparisons test, was used for comparing more than two conditions.

### Analysis of fluorescence intensities of transfected constructs

Vim-mCh-SspB, Vim-mCh-FKBP, FRB-BFP-GCN4-ppKin14, iLID-GFP-GCN4-ppKin14, KIF5A-mVenus-iLID, and HA-KIF5A-FRB (stained with anti-HA antibody) fluorescence intensities were quantified by defining ROI around individual cells, including transfected and non-transfected neighboring cells, and a background region for normalization. The mean gray value was measured for each ROI using the following equation:

$$\text{Normalized intensity} = \frac{(\text{transfected cell intensity} - \text{background})}{(\text{untransfected neighbour intensity} - \text{background})}$$

### Quantification of the fraction of cell area occupied by vimentin

To quantify the area occupied by vimentin, we thresholded the total vimentin intensity (as obtained by immunofluorescence staining) using the Li threshold algorithm in Fiji and then quantified the total vimentin area using the "Detect Particles" function. This value was then divided by the total cell area to calculate the cell area occupied by vimentin.

### Quantification of vimentin recovery at the cell periphery

Vimentin at the cell periphery was analyzed by drawing an ROI near the cell edge, at the cell edge furthest from the MTOC/vimentin cluster, between 10 and 15 μm in width, where the width was perpendicular to the cell edge. ROI was adjusted during cell movement, and the mean grey value was measured over time inside ROI. Background was subtracted from the measurement and then normalized to the first time point.

### Quantification of focal adhesion number

We utilized the ComDet plugin for Fiji (https://github.com/ekatrukha/ComDet) to detect the total number of focal adhesions based on paxillin staining. Phalloidin staining was employed to help define the total cell area. The total number of focal adhesions was then normalized by dividing it by the cell area.

### Analysis of microtubule intensity and radiality

To quantify the intensity of tyrosinated tubulin, we used ROIs around transfected and non-transfected cells and background areas. The mean grey value for each ROI was measured and used in the following formula:

$$\text{Normalized intensity} = \frac{(\text{transfected cell intensity} - \text{background})}{(\text{untransfected neighbour intensity} - \text{background})}$$

To quantify the total intensity of the radial and non-radial microtubules, we used the radiality map plugin (https://github.com/UU-cellbiology/radialitymap). After defining radial and non-radial microtubules, we calculated the ratio by dividing the mean intensity of non-radial microtubules by the mean intensity of radial microtubules over the same cell area.

For the quantification of acetylated tubulin, the mean gray values observed after background subtraction were normalized to the average acetylated tubulin intensity in non-transfected cells.

### Analysis of vimentin and keratin colocalization

Manders' colocalization coefficients for keratin-8 and vimentin networks were calculated using the JACoP plugin in Fiji (Bolte and Cordelières, 2006). Before measuring colocalization, we manually thresholded the images to accurately define the keratin-8 and vimentin networks. Coefficients were calculated from ROIs with dimensions of 50.2 × 50.2 μm for COS-7 cells, 45.11 × 45.11 μm for U2OS cells, and 35.03 × 35.03 μm for HeLa cells, corresponding to their average cell size. The Kruskal–Wallis test with Dunn's multiple comparisons test was used to compare the colocalization coefficients of keratin-8 to vimentin for the different conditions.

### Analysis of perinuclear to peripheral ER/vimentin intensity

A circle with a diameter of 27.9 μm was drawn around the nucleus/perinuclear cloud to quantify fluorescence intensity at the perinuclear area. A mask of the cell was created with the perinuclear area removed to quantify the fluorescence intensity at the cell periphery. A region outside the cell was drawn to quantify background fluorescence. The ratio of intensity at the perinuclear region to intensity at the cell periphery was then calculated as follows:

$$\text{Ratio perinuclear region to cell periphery} = \frac{\text{perinuclear intensity} - \text{background}}{\text{periphery intensity} - \text{background}}$$

### Analysis of ER and tagged vimentin intensity during localized vimentin pulling

The localized opto-vimentin pulling experiment in cells co-transfected with Halo-KDEL was analyzed by quantifying the

mean grey value of Vim-mCh-SspB and Halo-KDEL inside the blue light (488 nm) illuminated region over time. Background was subtracted, and values were normalized to the first time point.

### Analysis of lysosome and mitochondria distribution

Distributions of mitochondria and lysosomes were analyzed using LAMTOR4 and cytochrome C staining, respectively. The ComDet plugin was used to detect particles, with the cell center defined at the MTOC. A radius macro was used to calculate the distance of each particle from the cell (https://gist.github.com/ekatrukha/105553627f1bee01367faae153bfe5c0).

From these distance measurements, we determined the fraction of particles located in the cell periphery. The peripheral region was defined as an area beyond a circle with a 13.81-μm radius from the MTOC, based on the average size of the perinuclear cloud for U2OS cells. Particles outside this circle were classified as being in the peripheral area of the cell.

### Online supplemental material

Fig. S1 illustrates the motor recruitment to the filaments induced by the rapalog system and repositioning of the vimentin network in U2OS cells. Fig. S2 illustrates the optogenetic recruitment of the minus end–directed motor and vimentin pulling by plus end–directed kinesin. Fig. S3 demonstrates that the KIF5A overexpression reduces the abundance of acetylated microtubules. Fig. S4 depicts the endogenous vimentin and keratin-8 filament networks in COS-7, U2OS, and HeLa cells. Fig. S5 shows a representative force–distance curve for cell stiffness measurements. Video 1 shows the rapalog-induced vimentin repositioning by minus end–directed kinesin. Videos 2 and 3 demonstrate the rapalog-induced vimentin repositioning by plus end–directed kinesin. Video 4 illustrates the light-induced vimentin repositioning and recovery. Video 5 shows the effect of light-induced vimentin repositioning on the ER in the whole cell and a zoomed region. Videos 6 and 7 show the effects of local light-induced vimentin repositioning on the ER in a control cell and an RNF26 knockout cell, respectively. Video 8 demonstrates the effect of local light-induced vimentin repositioning on mitochondria. Video 9 shows the effect of light-induced vimentin repositioning on lysosomes in the whole cell and a zoomed region. SourceDataSF4 contains original uncropped gels for Fig. S4 A.

### Data availability

The data that support the conclusions are available in the manuscript; the original fluorescence microscopy datasets are available upon request to A. Akhmanova. Scripts used for data analysis are available at https://github.com/ekatrukha/ComDet; https://github.com/UU-cellbiology/radialitymap; https://gist.github.com/ekatrukha/105553627f1bee01367faae153bfe5c0.

## Acknowledgments

We thank L. Kapitein and G.G. Farías (Utrecht University) for providing materials, I. Berlin (Leiden University Medical Center) for sharing RNF26 knockout U2OS cells, and Eugene Katrukha (Utrecht University) for technical support and guidance on image analysis.

This work was supported by the Netherlands Organization for Scientific Research (NWO) project ECHO 711018004 and open competition grant OCENW.M20.054, EMBO long-term fellowship ALTF 261-2019, Gravitation programme IMAGINE! (project number 24.005.009), the European Research Council Synergy grant PushingCell (project number 101071793) to A. Akhmanova, and a grant of NWO Talent Programme (project number VI.C.182.004) to G.H. Koenderink.

Author contributions: M. Pasolli: conceptualization, data curation, formal analysis, investigation, methodology, software, validation, visualization, and writing—original draft, review, and editing. J.C.M. Meiring: conceptualization, data curation, formal analysis, investigation, methodology, project administration, resources, validation, visualization, and writing—original draft, review, and editing. J.P. Conboy: formal analysis, investigation, validation, visualization, and writing—original draft, review, and editing. G.H. Koenderink: data curation, formal analysis, funding acquisition, investigation, methodology, resources, software, supervision, validation, visualization, and writing—review and editing. A. Akhmanova: conceptualization, funding acquisition, project administration, supervision, and writing—original draft, review, and editing.

Disclosures: The authors declare no competing interests exist.

Submitted: 1 April 2025

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

# Supplemental material

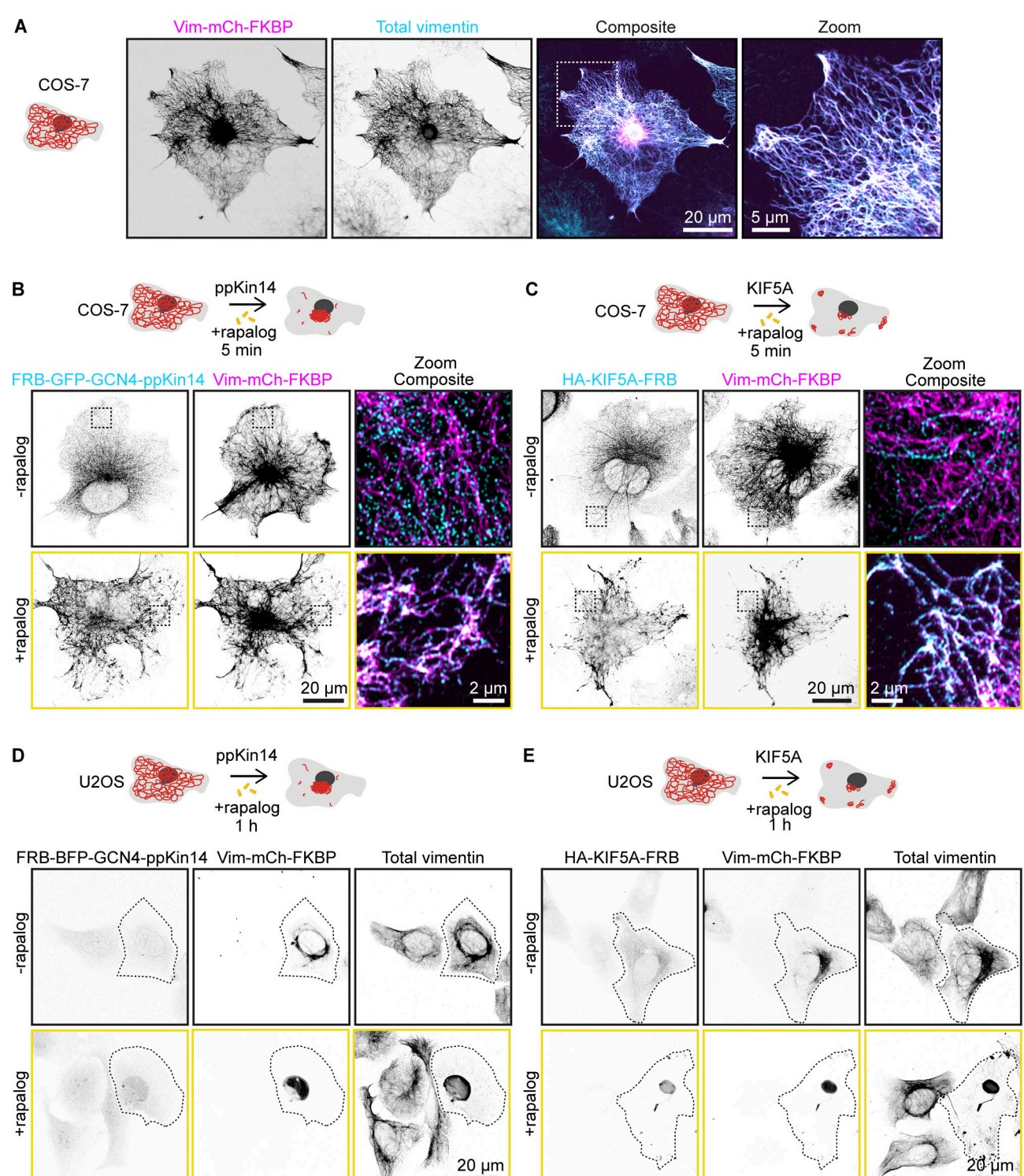

Figure S1. **Motor recruitment to the filaments with the rapalog system and repositioning of the vimentin network in U2OS cells. (A)** Fluorescence images of COS-7 cells expressing Vim-mCh-FKBP. The overexpressed vimentin is visualized by mCherry fluorescence, while total vimentin intensity is detected via anti-vimentin immunostaining. The dashed box indicates the region of the cell enlarged in the zoom image. Images were captured using Airyscan microscopy. **(B and C)** Fluorescence images of COS-7 cells co-transfected with Vim-mCh-FKBP and either FRB-GFP-GCN4-ppKin14 (B) or HA-KIF5A-FRB (C), with or without 5-min rapalog treatment. The top panels show untreated cells, while the bottom panels display rapalog-treated cells. FRB-GFP-GCN4-ppKin14 and HA-KIF5A-FRB are detected using anti-GFP and anti-HA antibodies respectively. Images were captured using Airyscan microscopy. **(D and E)** Representative fluorescence images of U2OS cells co-transfected with Vim-mCh-FKBP and FRB fusions of the indicated motors, with or without 1 h of rapalog treatment. Transfected cells are outlined with dashed lines, and non-transfected cells serve as controls. Anti-vimentin and anti-HA antibodies detect total vimentin and HA-KIF5A-FRB, respectively.

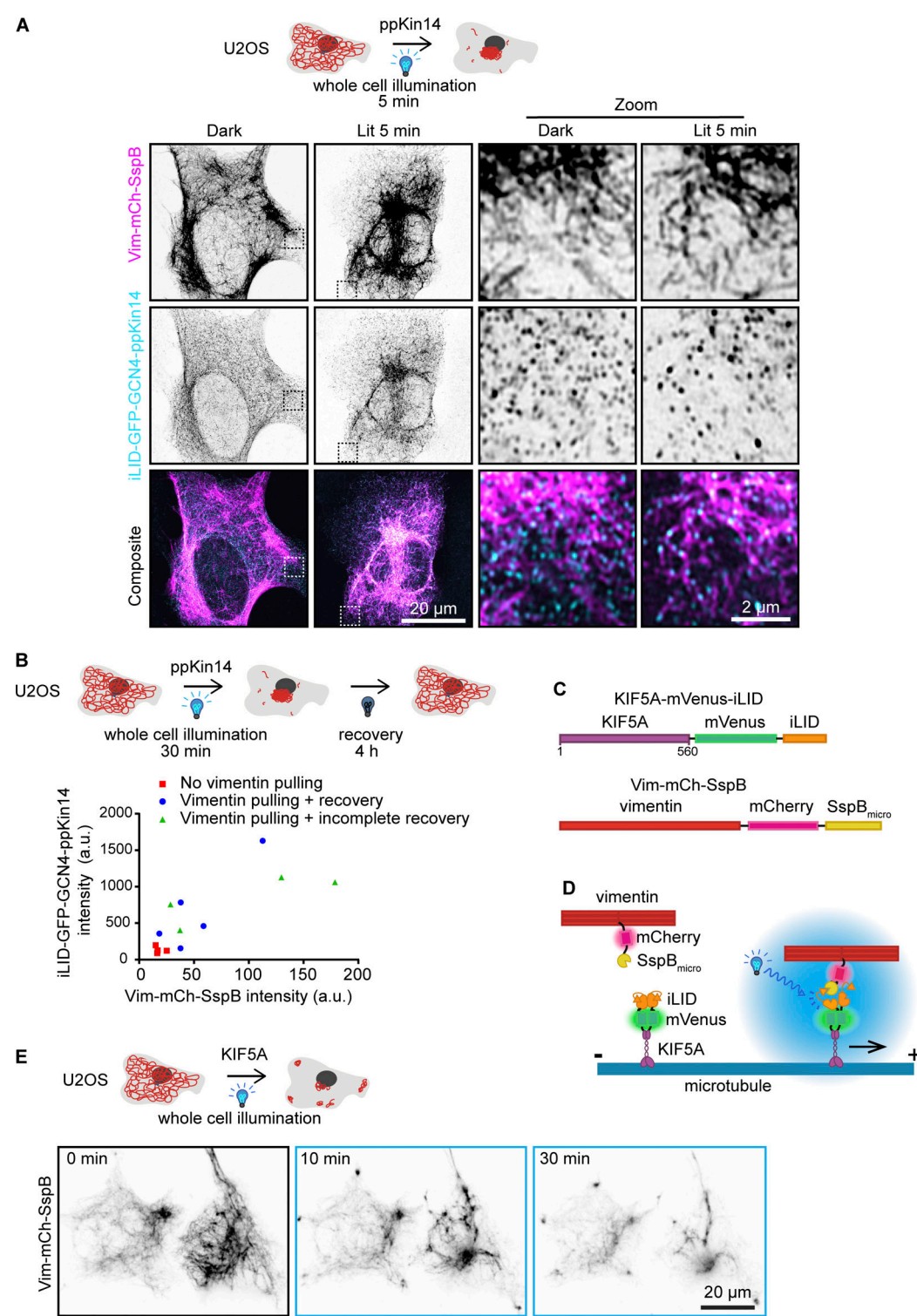

Figure S2. **Optogenetic minus end–directed motor recruitment and vimentin pulling by plus end–directed kinesin. (A)** U2OS cells co-transfected with Vim-mCh-SspB and iLID-GFP-GCN4-ppKin14 were either fixed in a dark room (DARK) or exposed to 5 min of blue light (LIT) prior to fixation and staining for anti-GFP. Dashed box shows the region of the cell enlarged in the zoom panel. Images were captured using Airyscan confocal microscopy. **(B)** U2OS cells co-transfected with Vim-mCh-SspB and iLID-GFP-GCN4-ppKin14 and imaged live using spinning disc confocal microscopy. Cells were pulsed with 488 nm light for 30 min over the entire cell and then allowed to recover their vimentin distribution in the absence of blue light stimulation for 4 h. The graph shows quantification of construct expression levels based on fluorescence at the first frame, with background subtracted. Cells were categorized either as having no vimentin clustering (red square), vimentin pulling with recovery of vimentin spreading after 4 h (blue circle), or vimentin pulling without complete recovery of vimentin spreading after 4 h (green triangle). **(C and D)** Schematic overview of constructs for optogenetic plus end–directed vimentin pulling. **(E)** U2OS cells co-transfected with Vim-mCh-SspB and KIF5A-mVenus-iLID show vimentin relocalization to clusters in the cell periphery and at the cell center upon blue light activation. Live-cell imaging stills are shown before blue light activation (0 min) and 10 and 30 min after 488-nm pulsing.

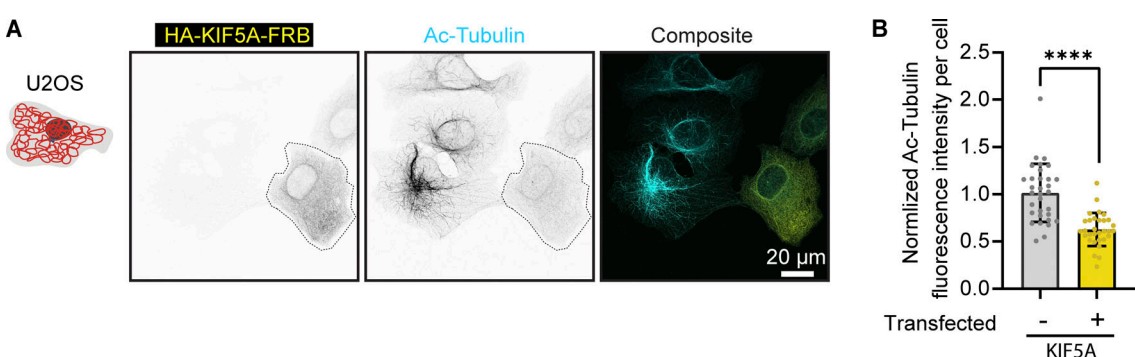

Figure S3.    **KIF5A overexpression diminishes the abundance of acetylated microtubules. (A)** Representative images of U2OS cells, either untransfected or transfected with HA-KIF5A-FRB, stained for acetylated microtubules (Ac-tubulin). The transfected cell is outlined with a dashed line. **(B)** Quantification of normalized acetylated microtubule intensity in untransfected U2OS cells and HA-KIF5A-FRB–expressing cells. A total of 31–32 cells were analyzed across three independent experiments. Data are presented as mean ± SD, with individual cell measurements shown as dots. ****, P < 0.0001 based on Mann–Whitney statistical analysis.

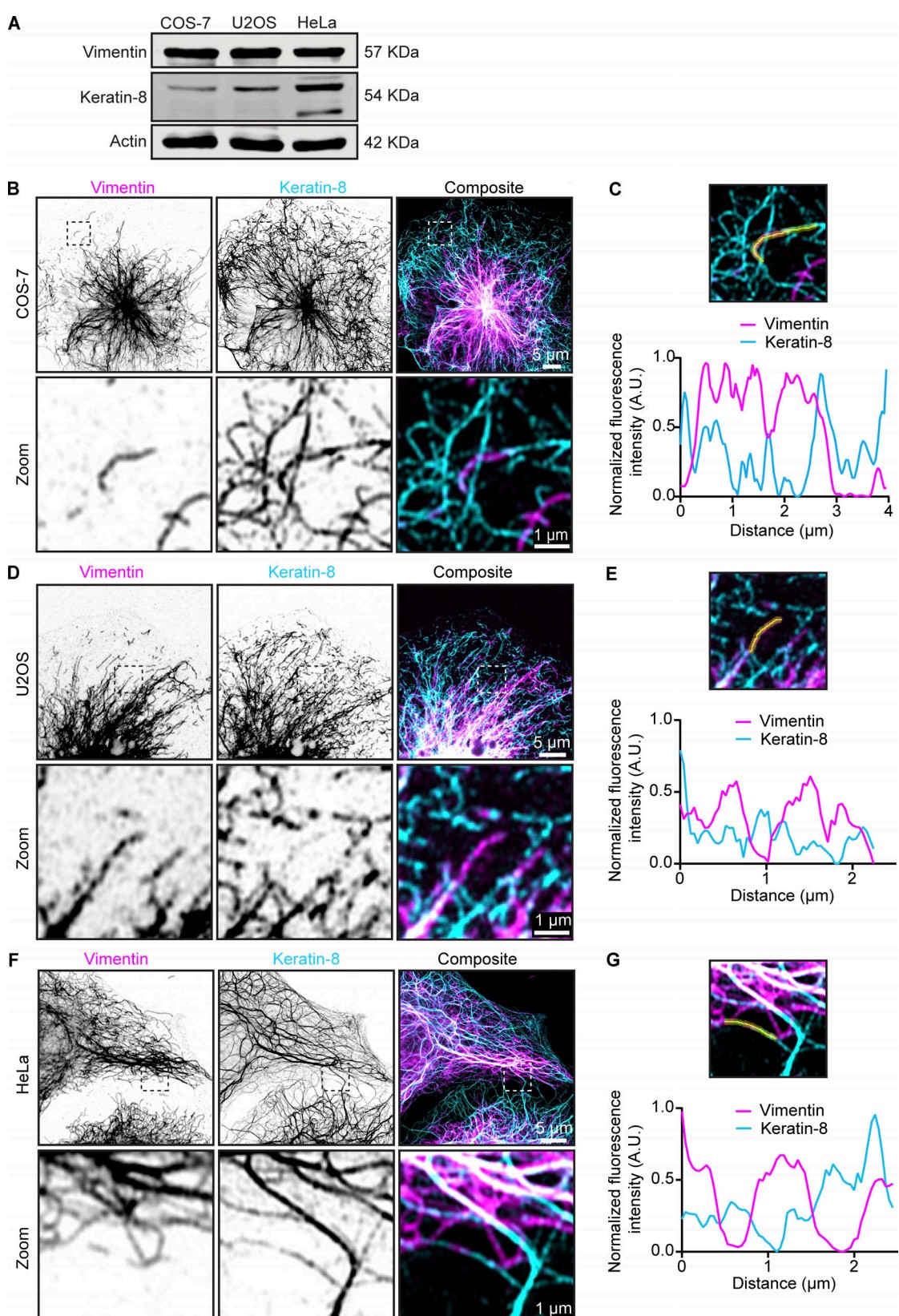

Figure S4.  **Endogenous vimentin and keratin-8 filament networks in COS-7, U2OS, and HeLa cells. (A)** Expression levels of vimentin and keratin-8 in COS-7, U2OS, and HeLa cells were analyzed by western blotting. **(B–G)** Airyscan high-resolution images of vimentin and keratin-8 networks in COS-7 (B), U2OS (D), and HeLa cells (F), detected by staining with anti-vimentin and anti–keratin-8 antibodies, respectively. Zoomed-in views (5 × 5 µm), highlighted by the dashed box, showing the region of the cell enlarged in the zoom panel and intensity profiles for vimentin and keratin-8 along the indicated lines in COS-7 (C), U2OS (E), and HeLa (G) cells. Source data are available for this figure: SourceData FS4.

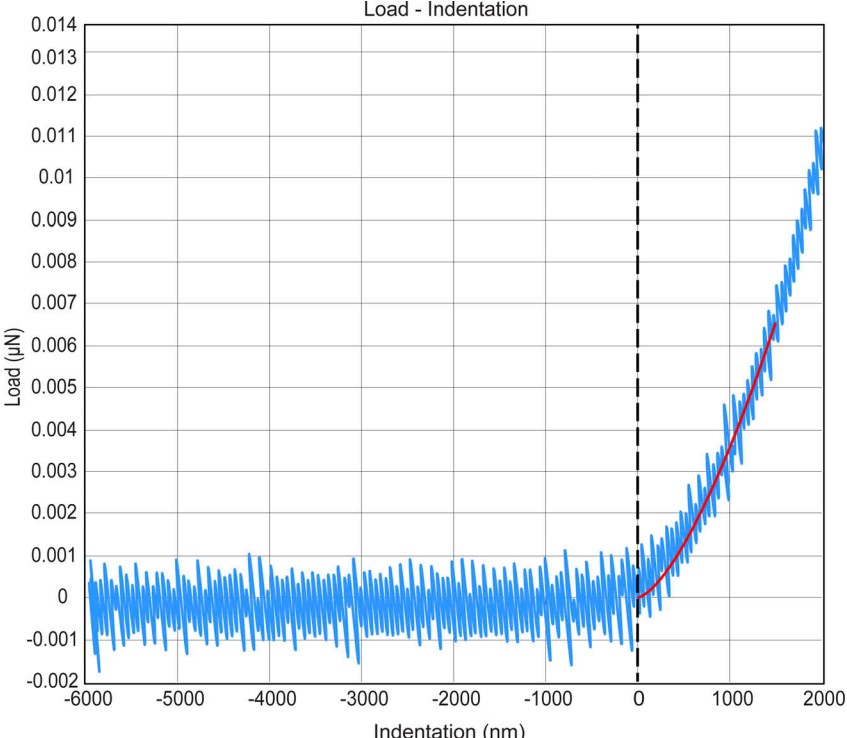

**Figure S5. A force–distance curve for cell stiffness measurements.** An example force curve used to determine the Young's modulus of the cells. The raw data are shown in blue, and the Hertz model fit is the red line. Fitting was restricted to the first 1.5 μm of indentation, as shown in the figure.

Video 1. **Rapalog-induced vimentin repositioning by minus end–directed kinesin.** COS-7 cell co-expressing Vim-mCh-FKBP and FRB-BFP-GCN4-ppKin14. Imaging was conducted for a total of 70 min; rapalog was added at the 10-min mark. Time is shown in hh:mm:ss.

Video 2. **Rapalog-induced vimentin repositioning by plus end–directed kinesin.** COS-7 cell co-expressing Vim-mCh-FKBP and HA-KIF5A-FRB. Total imaging time: 70 min; rapalog is added at the 10-min mark. Time is shown in hh:mm:ss.

Video 3. **Rapalog-induced vimentin repositioning by plus end–directed kinesin.** COS-7 cell co-expressing Vim-mCh-FKBP and HA-KIF5A-FRB. The movie starts from the moment the rapalog is added. Time is shown in hh:mm:ss.

Video 4. **Light-induced vimentin repositioning and recovery.** U2OS cell co-expressing Vim-mCh-SspB and iLID-GFP-GCN4-ppKin14 was exposed to 1 s of whole-cell blue light illumination every 30 s for a total of 30 min; a blue dot is shown in the top right corner during blue light illumination. The cell was subsequently imaged for another 4 h without blue light activation to allow vimentin distribution to recover. Time is shown in hh:mm.

Video 5. **Effect of light-induced vimentin repositioning on ER.** U2OS cells co-transfected with Vim-mCh-SspB, iLID-GFP-GCN4-ppKin14, and Halo-KDEL (ER marker) were first imaged for 5 min without blue light and then imaged for another 25 min with 500 ms of whole-cell blue light illumination every 4 s; the period of blue light pulsing is indicated by a blue dot in the top right corner. Time is shown in mm:ss.

Video 6. **Effect of local light-induced vimentin repositioning on ER in a control cell.** WT U2OS cells co-transfected with Vim-mCh-SspB, iLID-GFP-GCN4-ppKin14, and Halo-KDEL (ER marker) were illuminated inside an ROI (indicated by a blue box) with one 113-ms pulse of blue light every 4 s; the period of blue light pulsing is indicated by a blue dot in the top right corner. Time is shown in mm:ss.

Video 7. **Effect of local light-induced vimentin repositioning on ER in an RNF26 knockout cell.** RNF26 KO U2OS cells co-transfected with Vim-mCh-SspB, iLID-GFP-GCN4-ppKin14, and Halo-KDEL (ER marker) were illuminated inside an ROI (indicated by a blue box) with one 113-ms pulse of blue light every 4 s; the period of blue light pulsing is indicated by a blue dot in the top right corner. Time is shown in mm:ss. KO, knockout.

Video 8. **Effect of local light-induced vimentin repositioning on mitochondria.** U2OS cells co-transfected with Vim-mCh-SspB (magenta), iLID-GFP-GCN4-ppKin14, and Halo-MitoTag (cyan, mitochondria marker) were illuminated inside an ROI (indicated by a blue box) with one 113-ms pulse of blue light every 4 s; the period of blue light pulsing is indicated by a blue dot in the top right corner. Time is shown in mm:ss.

Video 9. **Effect of light-induced vimentin repositioning on lysosomes.** U2OS cells co-transfected with Vim-mCh-SspB (cyan), iLID-GFP-GCN4-ppKin14, and LAMP1-Halo (yellow, lysosome marker) were exposed to 500 ms of blue light illumination every 4 s; the period of blue light pulsing is indicated by a blue dot in the top right corner. Time is shown in mm:ss.

