## [Peer Review File · The Journal of Cell Biology]

Optogenetic and chemical genetic tools for rapid repositioning of vimentin intermediate filaments

Milena Pasolli, Joyce Meiring, James Conboy, Gijsje Koenderink, and Anna Akhmanova

Corresponding Author(s): Anna Akhmanova, Utrecht University

Review Timeline:

Submission Date:	2025-04-01
Editorial Decision:	2025-05-24
Revision Received:	2025-05-30

Monitoring Editor: Sandrine Etienne-Manneville

Scientific Editor: Dan Simon

Transaction Report:

DOI: <https://doi.org/10.1083/jcb.202504004>

Revision 0

Review #1

1. Evidence, reproducibility and clarity:

Evidence, reproducibility and clarity (Required)

****Summary**** The manuscript is well written, with excellent explanation and documentation of experimental approaches. All conclusions are well supported by the data. The discussion is balanced and appropriate. The data, including images and movies, are of high quality and beautifully presented. The experimental design and analysis, including quantification of parameters in the images, is rigorous. Additional rigor is provided by comparing different cell types. The rapalog and iLID dimerization strategies have been described previously, as has their use to recruit kinesin motors to membranous organelles. However, this is the first application of these strategies to recruit motors to intermediate filaments. The evidence that vimentin filaments can be redistributed locally is clear and convincing and offers appealing potential for future experimentation. The redistribution was not fully reversible in all cells, but this is not surprising given the entanglement that must result from the action of motors along the length of these long flexible polymers.

In terms of the biology of intermediate filaments, the authors show that vimentin redistribution had negligible effect on microtubule or F-actin organization, cell area, or the number of focal adhesions. Depletion of vimentin filaments locally reduced cell stiffness. Both ER and mitochondria segregated with vimentin filaments, but not lysosomes. These findings are consistent with published reports (e.g. comparing vimentin null and wildtype cell lines), but the acute and reversible nature of the motor recruitment strategy is a more elegant experimental approach, and the selectivity of the observed effects is evidence of its specificity. It is interesting that the ER network segregated with vimentin even in the absence of RNF26. While this is not explored further, it points to the potential power of this motor recruitment strategy for future studies on intermediate filament interactions.

The following are some major and minor issues, which should all be easy for the authors to address.

****Major Comments:****

- Fig. S1 shows that the Vim-mCherry-FKBP construct coassembles with endogenous vimentin,

but similar data for the iLID constructs appears to be lacking. I would like to see data demonstrating the incorporation of the Vim-mCherry-SspB constructs into the vimentin filaments. This should include high magnification images of single filaments in the cytoplasm of the cells.

- The authors do not discuss the density of motor recruitment along the filaments. To address this, I'd like to see images showing the extent of recruitment of motors to the filaments using the rapalog and LID strategies. This should include high magnification images of single filaments in the cytoplasm of the cells.

- For the experiments on vimentin and keratin organization, the authors do not explain that these proteins form distinct networks and do not coassemble. The authors should show this in the cell types examined. This should also be explained explicitly in the body of the manuscript, though the data could be placed in the supplementary data. This is important because many intermediate filaments can coassemble freely, and coassembled proteins would be expected to segregate together.

****Minor Comments:****

- The authors refer to selecting cells within an "optimized expression range" for their transiently expressed recombinant proteins. They should state the proportion of the cells that met this criterion in their transient transfection experiments as this is important information for other researchers that might wish to use this approach in their own studies.

- In Fig. 1F there should be a statistical comparison between cells transfected with the Kin14 construct and control (untransfected) cells in the absence of rapalog

- In Fig. 1G there should be a statistical comparison between cells expressing Kin14 and KIF5A in the absence of rapalog

- The depletion of the ER network in the cell periphery is not evident in Fig. 7B, though the perinuclear accumulation is evident. Perhaps the authors could select another example or explain to the reader what exactly to look for in these images.

- In Fig. 7C, the intensity of the mCherry declines markedly over time. This is presumably due to photobleaching but should be explained in the legend.

****Referees cross-commenting****

This session contains comments of Reviewer 1 and Reviewer 2

Reviewer 1:

I don't understand what Reviewer 2 means by "A major shortcoming is the unclear narrative, what do the authors want to present? This aspect requires significant attention." I found the narrative, purpose and conclusions of this study very clear to me. I also do not understand Reviewer 2's concern with the abstract. I re-read it and it still seems very clear and appropriate to me. For example, the authors state "Here, we present tools that allow rapid manipulation of vimentin IFs in the whole cytoplasm or within specific subcellular regions by inducibly coupling them to microtubule motors, either pharmacologically or using light". This seems clear and correct to me. It would be helpful if Reviewer 2 could point to specific language and explain why it is problematic.

Reviewer 2:

The strength of this paper is clearly the strong methods development and I find this aspect very intriguing and attractive. There is an imbalance in the narrative presenting on one hand the method and on the other hand presenting concrete research results. In my view, although interesting, the different experimental results serve more as proof-of-concept and they should not be presented as bona fide evidence of an existing or lacking bilateral interrelationship.

Indeed, the cited sentence makes sense: "Here, we present tools that allow rapid manipulation of vimentin IFs in the whole cytoplasm or within specific subcellular regions by inducibly coupling them to microtubule motors, either pharmacologically or using light." as it features the methods aspect of the paper. However, the following sentences: "Perinuclear clustering of vimentin had no strong effect on the actin or microtubule organization, cell spreading, and focal adhesions, but reduced cell stiffness. Mitochondria and endoplasmic reticulum sheets were repositioned together with vimentin, whereas lysosomes were only briefly repositioned and rapidly regained their normal distribution. Keratin was displaced along with vimentin in some cell lines but remained intact in others. " embraces everything from actin to microtubules to cell spreading to focal adhesions to cell stiffness to mitochondrial function to lysosomes to interactions with other IF family members etc. This gives the impression that the authors want to make claims on how vimentin affects or does not affect these cellular functions and structures and once just cannot make such sweeping claims with so little evidence. With the experimental setting included, non of these claims can be really made without rigidly examining each and every interaction (which has been done separately for many of these bilateral interactions during the past 20 years or so).

Hence, it should be made clear that these observations are used and mentioned as proof of concept that the tool is working, not as evidence that this or that interaction takes place or does not take place. As I indicated in my review, such claims on any of these bilateral interactions would require a lot more evidence to be properly substantiated.

My comment is to be regarded as a positive one. If I would judge the paper based on how one could interpret the abstract and the text regarding, for example, that vimentin does not affect focal adhesions but changes cellular stiffness, my review would be significantly more stringent. However, I would really like to see this paper being published, but the claims on revealing new vimentin functions or disproving earlier observations based on these very limited data are just not sufficiently substantiated to be acceptable. Hence, I urge the authors to adjust the narrative to be clear on the methods development, which is also the focus of the title. I believe this is a justified recommendation and also, overall, a fair shake of the study and a constructive approach on how to publish this manuscript without extensive experiments.

Reviewer 1:

I thank Reviewer 2 for this explanation. I do understand their point. However, while not the end of the story, I do feel the authors' data are a bit more than just a proof of principle and do offer important insights into the biology which the field will need to grapple with. Each graph includes measurements on dozens of cells from multiple experiments and there is clearly selectivity to what segregates with the vimentin filaments and what does not. I would just ask the authors to be a bit more nuanced in their interpretation and conclusions about the biology to address Reviewer 2's concerns.

Reviewer 2:

That sounds like a fair assessment. Main thing is that this data is presented in a balanced way, with emphasis on the model development. Some of the presented data are in contradiction with quite established concepts by several researchers and the data presented here does not substantiate a paradigm shift. Regardless of this, some pieces of the data are intriguing, for example, the live cell imaging.

2. Significance:

Significance (Required)

****Summary:**** The authors show that chemical-induced and light-induced dimerization

strategies can be used to recruit microtubule motors to vimentin filaments, allowing rapid and reversible experimental manipulation of vimentin filament organization either locally or globally in cells. These strategies provide an experimental approach for investigating the physical interaction of intermediate filaments with organelles and other cytoskeletal component, as well as a method for probing the role of intermediate filaments in cell mechanics, cytoskeletal dynamics, etc. This is a technical improvement over previous experimental strategies, which have relied largely on chronic manipulation such as global disassembly or genetic deletion of intermediate filaments, e.g. comparison of vimentin null and wild type cells.

The principal weakness of this study is that it offers limited insight into intermediate filament biology. As such, it might be most appropriate for a tools or techniques section of a journal. The dimerization strategies have been reported previously, so that is not new, but the application to intermediate filaments is novel.

****Audience:**** This paper will be of interest to cell biologists who study cytoskeletal interactions, particularly the interaction of intermediate filaments with other cellular organelles or cytoskeletal polymers, or the role of intermediate filaments in cellular mechanics.

****Reviewer Expertise**** This reviewer has expertise on the cytoskeleton, cytoskeletal dynamics, and intracellular transport including intermediate filament biology.

3. How much time do you estimate the authors will need to complete the suggested revisions:

Estimated time to Complete Revisions (Required)

(Decision Recommendation)

Less than 1 month

4. Review Commons values the work of reviewers and encourages them to get credit for their work. Select 'Yes' below to register your reviewing activity at Web of Science Reviewer Recognition Service (formerly Publons); note that the content of your review will not be visible on Web of Science.

No

Review #2

1. Evidence, reproducibility and clarity:

Evidence, reproducibility and clarity (Required)

****Summary:**** The manuscript presents a novel methodology for acute manipulation of vimentin intermediate filaments (IFs) using chemical genetic and optogenetic tools. By recruiting microtubule-based motors to vimentin via inducible dimerization systems, the authors achieve precise temporal and spatial control over vimentin distribution. Apart from the significant advancement in terms of methods development, key findings include:

- Vimentin's role in organelle positioning: Mitochondria and ER are repositioned with vimentin, while lysosomes are less dependent on its organization.
- Cytoskeletal interactions: Vimentin clustering minimally impacts actin and microtubule networks in the short term.
- Cell stiffness: Vimentin repositioning reduces cell stiffness, indicating its significant role in cellular mechanics.
- Cell-type-specific keratin interactions: The study highlights diverse interactions between vimentin and keratin-8 across cell lines.

The study demonstrates methodological advancements enabling rapid vimentin manipulation and provides insights into vimentin's interactions with cellular structures.

A major shortcoming is the unclear narrative, what do the authors want to present? This aspect requires significant attention.

General Comments and Overall Assessment

The manuscript represents an interesting contribution to the cytoskeletal field, addressing limitations of long-term perturbation methods. The tools developed are innovative, allowing controlled and reversible vimentin reorganization with minimal off-target effects. The findings are robust and provide important insights into the role of vimentin in cellular mechanics and organelle positioning.

Strengths:

Methodological novelty with broad applicability - this is the most exciting aspect.

Comprehensive validation of the tools in multiple cell lines.

Clear differentiation between vimentin's short- and long-term roles.

Addressing gaps in understanding vimentin-organelle interactions.

Limitations:

- The manuscript is a little bit all over the place. While the method development is clear, the manuscript makes claims way beyond the method development. The message and narrative needs to be improved, and in the respect the whole structure needs an overhaul.

- Unclear how much the differences in expression levels impact results and reproducibility.

- Would be good to discuss some findings that are specific to a given experimental cell line. How generalizable are these results?

****Major Comments****

Evidence and Claims:

- While the methodological aspect is very strong the balance between presenting a novel method and presenting specific cell biological findings needs to be improved. Now it is quite unclear what the manuscript wants to present.

- The abstract needs a complete overhaul. From reading the abstract, it is not clear what the manuscript wants to present.

Regarding the research findings there are a number of things for the authors to consider. Since the methods aspect is, in the eyes of this reviewer, in focus, I have not stringently assessed the experimental findings. Hence, the comments below are things to be considered in order to make

the findings related to IF research stronger:

- Cell-specific keratin interactions: The manuscript could benefit from some further validation of the physical interactions between vimentin and keratin-8 across different cell types.
- Impact on microtubules: The disorganization of stable microtubules in cells expressing KIF5A was attributed to overexpression effects. It would be helpful to include additional controls, such as expressing KIF5A without vimentin constructs, to confirm this claim.
- ER-vimentin linkages: The observation that ER-vimentin interactions persist in RNF26 knockout cells is intriguing. The manuscript would benefit from a discussion on possible candidates for alternative linkers.
- Construct variability: Do the authors have some data on how much Expression level differences significantly affect the outcomes (e.g., incomplete recovery)?

2. Significance:

Significance (Required)

General Assessment: The study represents a significant technical advance in the study of cytoskeletal dynamics. The tools developed address critical limitations of traditional vimentin perturbation methods, allowing for spatiotemporally precise manipulation without long-term effects on gene expression or signaling pathways.

Novelty:

This is, to my knowledge, the first demonstration of reversible and acute vimentin repositioning using optogenetics.

The study extends understanding of vimentin's short-term mechanical and organizational roles, distinguishing them from compensatory effects observed in knockdown models.

Audience and Impact: The manuscript will appeal to researchers in cytoskeletal dynamics, cell mechanics, and organelle biology. The tools have broader applicability in studying other cytoskeletal systems and could inspire translational applications, such as investigating the role of vimentin in cancer or fibrosis.

The reference list provide a relatively representative selection of articles relevant for the article. However, the authors may consider whether there could be relevant information in the relatively recent special edition of Current Opinion in Cell Biology, which focused on IFs, specially featuring vimentin <https://www.sciencedirect.com/special-issue/10TFHK2QCKW>

Field of Expertise

I specialize in cell biology, intermediate filaments, post-translational modifications, cytoskeletal dynamics, and advanced microscopy techniques.

3. How much time do you estimate the authors will need to complete the suggested revisions:

Estimated time to Complete Revisions (Required)

(Decision Recommendation)

Less than 1 month

4. Review Commons values the work of reviewers and encourages them to get credit for their work. Select 'Yes' below to register your reviewing activity at Web of Science Reviewer Recognition Service (formerly Publons); note that the content of your review will not be visible on Web of Science.

Yes

Review #3

1. Evidence, reproducibility and clarity:

Evidence, reproducibility and clarity (Required)

****Summary:****

This is an excellent paper describing the use of chemical and light-induced heterodimerization of microtubule-based motors to rapidly disrupt the distribution of the vimentin cytoskeletal network. Rapid clustering of vimentin did not significantly affect the microtubule or actin

networks, cell spreading or focal adhesions. Other organelles were repositioned together with vimentin. Interestingly, in some cell lines, keratin networks were displaced along with vimentin while in other cells they were not.

****Major comments:****

The conclusions are well supported by the data presented and appropriate controls are included.

****Optional comments:****

1. The authors should expand on why they think the plus end directed KIF5A gives such a strong localization of vimentin to the perinuclear area.

2. Consideration should be given to the idea that the pulling of ER and mitochondria along with the vimentin could be due to trapping of these organelles within the vimentin matrix and not necessarily due to direct interactions. Such reasoning could explain the transient localization of lysosomes with the center aggregate since lysosomes are generally not thought to significantly bind to vimentin networks.

2. Significance:

Significance (Required)

This study describes some valuable tools that should be useful to cell biologists interested in determining the role of the cytoskeleton and possibly other organelles in a variety of cellular contexts. It overcomes some of the existing shortcomings of the pharmacological reagents currently available for studying intermediate filament biology and will provide a useful adjunct to other more long-term manipulations of the cytoskeleton. While much of the data presented confirm results obtained by other methods, this is a significant technical advance as it provides a short time scale, and in one instance, reversible manipulation of the cytoskeleton.

3. How much time do you estimate the authors will need to complete the suggested revisions:

Estimated time to Complete Revisions (Required)

(Decision Recommendation)

Less than 1 month

4. Review Commons values the work of reviewers and encourages them to get credit for their

work. Select 'Yes' below to register your reviewing activity at Web of Science Reviewer Recognition Service (formerly Publons); note that the content of your review will not be visible on Web of Science.

Yes

Manuscript number: RC-2024-02801

Corresponding author(s): Anna Akhmanova

1. General Statements

In this paper, we describe a new approach to rapidly manipulate vimentin intermediate filaments using pharmacological and optogenetic tools. While excellent pharmacological tools exist to trigger disassembly of actin and microtubules, functional studies of intermediate filaments mostly rely on their knockout, depletion or mutation. Here, we introduce and validate a method to rapidly (within minutes) clear the cytoplasm of vimentin by inducibly coupling it to microtubule motors. We show that this approach can be used to study vimentin cross-talk with other cytoskeletal filaments, membrane organelles and to rapidly manipulate cell stiffness. As a novel biological observation, we provide evidence of a varied degree of co-assembly of vimentin and keratin filaments in different cell lines using both super-resolution microscopy and co-displacement assays. Our tools will allow revisiting many important questions in the intermediate filament field. For example, they will allow to distinguish short-term and direct effects of intermediate filaments on cell polarity, adhesion and migration from their function in signaling and gene expression. We thus think that these tools will be of interest to a broad cell biological audience.

The paper has been reviewed by three reviewers, and we sincerely thank them for their constructive comments, which we have addressed in full by adding new data and by revising the text and the figures of the paper. The reviewers agreed that our approach is both novel and valuable and thus indeed represents “a significant technical advance”. Reviewers 1 and 3 raised only minor concerns, requesting some additional controls, clarifications and statistics, and we have fully addressed these comments. Reviewer 2 also had some minor comments which we have fully addressed. Furthermore, this reviewer indicated that we should adjust the writing of the manuscript, and we have done so in the revised version of the paper. Please note that the text below also includes the discussion between Reviewers 1 and 2 on the clarity of our narrative, and our responses to this discussion.

Reviewer #1 (Evidence, reproducibility and clarity (Required)):

SUMMARY: The manuscript is well written, with excellent explanation and documentation of experimental approaches. All conclusions are well supported by the data. The discussion is balanced and appropriate. The data, including images and movies, are of high quality and beautifully presented. The experimental design and analysis, including quantification of parameters in the images, is rigorous. Additional rigor is provided by comparing different cell types. The rapalog and iLID dimerization strategies have been described previously, as has

their use to recruit kinesin motors to membranous organelles. However, this is the first application of these strategies to recruit motors to intermediate filaments. The evidence that vimentin filaments can be redistributed locally is clear and convincing and offers appealing potential for future experimentation. The redistribution was not fully reversible in all cells, but this is not surprising given the entanglement that must result from the action of motors along the length of these long flexible polymers.

In terms of the biology of intermediate filaments, the authors show that vimentin redistribution had negligible effect on microtubule or F-actin organization, cell area, or the number of focal adhesions. Depletion of vimentin filaments locally reduced cell stiffness. Both ER and mitochondria segregated with vimentin filaments, but not lysosomes. These findings are consistent with published reports (e.g. comparing vimentin null and wildtype cell lines), but the acute and reversible nature of the motor recruitment strategy is a more elegant experimental approach, and the selectivity of the observed effects is evidence of its specificity. It is interesting that the ER network segregated with vimentin even in the absence of RNF26. While this is not explored further, it points to the potential power of this motor recruitment strategy for future studies on intermediate filament interactions.

The following are some major and minor issues, which should all be easy for the authors to address.

MAJOR COMMENTS:

- *Fig. S1 shows that the Vim-mCherry-FKBP construct coassembles with endogenous vimentin, but similar data for the iLID constructs appears to be lacking. I would like to see data demonstrating the incorporation of the Vim-mCherry-SspB constructs into the vimentin filaments. This should include high magnification images of single filaments in the cytoplasm of the cells.*

We have included a new Figure 2D, which illustrates the incorporation of the vimentin-mCherry-SspB construct into the vimentin network stained for endogenous vimentin.

- *The authors do not discuss the density of motor recruitment along the filaments. To address this, I'd like to see images showing the extent of recruitment of motors to the filaments using the rapalog and LID strategies. This should include high magnification images of single filaments in the cytoplasm of the cells.*

We have included new Figure S1B,C and Figure S2A, which illustrate the recruitment of kinesin motors to vimentin filaments upon induction with rapalog or light, respectively, by using super-resolution imaging with an Airyscan microscope. The motors were stained with antibodies against GFP. These data are discussed in the text, lines 126-132 and 165-168.

- *For the experiments on vimentin and keratin organization, the authors do not explain that these proteins form distinct networks and do not coassemble. The authors should show this in the cell types examined. This should also be explained explicitly in the body of the manuscript, though the data could be placed in the supplementary data. This is important because many*

intermediate filaments can coassemble freely, and coassembled proteins would be expected to segregate together.

To address this important comment, we have now included images of vimentin and keratin in the three studied cell types using super-resolution imaging, both for cells expressing vimentin constructs (updated Figure 5) and endogenous filament staining in untransfected cells (updated Figure S4). These images illustrate that vimentin and keratin mostly form distinct filaments in HeLa cells. However, we do observe some degree of co-assembly of vimentin and keratin in COS-7 and U2OS cells. We were really surprised by this observation as, to our knowledge, it has not been clearly documented in the literature. These data help to explain why vimentin pulling causes keratin co-clustering in COS-7 and U2OS cells. We note that in a study where kinesin-1 mediated transport of vimentin and keratin has been previously investigated by the Gelfand lab in RPE1 cells, the two networks also appear to overlap quite strongly (Robert et al, 2019, FASEB J). Since no super-resolution microscopy was performed in that study, potential co-assembly of keratin and vimentin filaments was not discussed. Colocalization and coprecipitation of vimentin and keratin have been also described by Velez-delValle et al. in epithelial cells (Sci Rep 2016). Cell type-specific co-assembly of keratin and vimentin would require more investigation, and we make no strong conclusions about it, but we think that our data illustrate the usefulness of our methodology to address the co-dependence of different types of intermediate filaments.

MINOR COMMENTS:

- *The authors refer to selecting cells within an "optimized expression range" for their transiently expressed recombinant proteins. They should state the proportion of the cells that met this criterion in their transient transfection experiments as this is important information for other researchers that might wish to use this approach in their own studies.*

These numbers are now included in lines 137 -142 and 173-176 of the revised paper. For the FRB-FKP system, ~50% of transfected cells could be used for analysis, for the light-induced system, ~40% were in the optimal range.

- *In Fig. 1F there should be a statistical comparison between cells transfected with the Kin14 construct and control (untransfected) cells in the absence of rapalog*

This comparison has been added.

- *In Fig. 1G there should be a statistical comparison between cells expressing Kin14 and KIF5A in the absence of rapalog.*

This comparison has been added.

- *The depletion of the ER network in the cell periphery is not evident in Fig. 7B, though the perinuclear accumulation is evident. Perhaps the authors could select another example or explain to the reader what exactly to look for in these images.*

We note that Figure 7B is a line scan of the image shown in Figure 7A. We assume that the reviewer meant Figure 7C, which is discussed in detail below.

• In Fig. 7C, the intensity of the mCherry declines markedly over time. This is presumably due to photobleaching but should be explained in the legend.

We have now improved Figure 7 by adding additional quantifications of ER and vimentin intensity and distribution in Figures 7D and E. We also extended the corresponding text (lines 288-297), which now reads; “Using the optogenetic tool, we observed that ER sheets and matrices, but not tubules, were pulled along with vimentin, confirming their previously described direct connections (Cremer et al., 2023) (black arrows, Figure 7C; Video S5). Most of the vimentin and ER repositioning occurred within approximately 10 minutes (Figure 7C, D, Video S5). While initially this resulted in a sparser tubular ER network at the cell periphery, over time, the network became denser, with smaller polygonal structures. This effect could also be observed in the ratio of perinuclear to peripheral intensity, where a subset of ER initially follows vimentin to the perinuclear region but then redistributes again towards the cell periphery (Figure 7D). It should be noted that while photobleaching of the ER channel was negligible, there was a 40% reduction in total Vim-mCh-SspB intensity over the course of the experiment due to photobleaching (Figure 7E).”

Reviewer #1 (Significance (Required)):

SUMMARY: *The authors show that chemical-induced and light-induced dimerization strategies can be used to recruit microtubule motors to vimentin filaments, allowing rapid and reversible experimental manipulation of vimentin filament organization either locally or globally in cells.*

These strategies provide an experimental approach for investigating the physical interaction of intermediate filaments with organelles and other cytoskeletal component, as well as a method for probing the role of intermediate filaments in cell mechanics, cytoskeletal dynamics, etc. This is a technical improvement over previous experimental strategies, which have relied largely on chronic manipulation such as global disassembly or genetic deletion of intermediate filaments, e.g. comparison of vimentin null and wild type cells.

The principal weakness of this study is that it offers limited insight into intermediate filament biology. As such, it might be most appropriate for a tools or techniques section of a journal. The dimerization strategies have been reported previously, so that is not new, but the application to intermediate filaments is novel.

We agree that our paper is primarily of technical nature and thus would be most appropriate for the tools and techniques section of a journal. We also agree that we used motor recruitment strategies that we and others have employed previously. However, we would like to emphasize that the demonstration that the tools work very well for intermediate filaments is entirely novel, as are the observations that these tools can be used to very rapidly alter cell stiffness or probe the links between intermediate filaments and organelles. Most importantly, the intermediate filament field currently lacks rapid specific manipulation strategies, and our tools will allow revisiting many important pending questions in the field. For example, they will allow to distinguish short-term and direct effects of intermediate filaments on cell polarity, adhesion and migration from their function in signaling and gene expression. We also report some new biology, such as evidence of some degree of co-assembly of vimentin and keratin.

Full Revision

AUDIENCE: This paper will be of interest to cell biologists who study cytoskeletal interactions, particularly the interaction of intermediate filaments with other cellular organelles or cytoskeletal polymers, or the role of intermediate filaments in cellular mechanics.

REVIEWER EXPERTISE: This reviewer has expertise on the cytoskeleton, cytoskeletal dynamics, and intracellular transport including intermediate filament biology.

Reviewer #2 (Evidence, reproducibility and clarity (Required)):

Summary: *The manuscript presents a novel methodology for acute manipulation of vimentin intermediate filaments (IFs) using chemical genetic and optogenetic tools. By recruiting microtubule-based motors to vimentin via inducible dimerization systems, the authors achieve precise temporal and spatial control over vimentin distribution. Apart from the significant advancement in terms of methods development, key findings include:*

** Vimentin's role in organelle positioning: Mitochondria and ER are repositioned with vimentin, while lysosomes are less dependent on its organization.*

** Cytoskeletal interactions: Vimentin clustering minimally impacts actin and microtubule networks in the short term.*

** Cell stiffness: Vimentin repositioning reduces cell stiffness, indicating its significant role in cellular mechanics.*

** Cell-type-specific keratin interactions: The study highlights diverse interactions between vimentin and keratin-8 across cell lines.*

The study demonstrates methodological advancements enabling rapid vimentin manipulation and provides insights into vimentin's interactions with cellular structures.

A major shortcoming is the unclear narrative, what do the authors want to present? This aspect requires significant attention.

As clarified below in the discussion between the two reviewers, by “unclear narrative” the reviewer meant that we should have provided a more balanced discussion of the insights that could be obtained using our new method compared to previously published literature, and we have modified our narrative accordingly.

General Comments and Overall Assessment

The manuscript represents an interesting contribution to the cytoskeletal field, addressing limitations of long-term perturbation methods. The tools developed are innovative, allowing controlled and reversible vimentin reorganization with minimal off-target effects. The findings are robust and provide important insights into the role of vimentin in cellular mechanics and organelle positioning.

Strengths:

Methodological novelty with broad applicability - this is the most exciting aspect.

Comprehensive validation of the tools in multiple cell lines.

Clear differentiation between vimentin's short- and long-term roles.

Addressing gaps in understanding vimentin-organelle interactions.

Limitations:

** The manuscript is a little bit all over the place. While the method development is clear, the manuscript makes claims way beyond the method development. The message and narrative needs to be improved, and in the respect the whole structure needs an overhaul.*

We have carefully modified the manuscript to avoid the impression that we make any claims that go beyond the immediate and quantifiable effects of vimentin repositioning on different cellular structures.

* Unclear how much the differences in expression levels impact results and reproducibility. Quantifications of expression levels and their discussion are included in Figures 1G-I, 2G-H, S2B and lines 137-142 and 173-176.

* Would be good to discuss some findings that are specific to a given experimental cell line. How generalizable are these results?
Cell line-specific findings concerned mostly the co-displacement of keratin together with vimentin, which occurred in COS-7 and U2OS cells but in HeLa cells. This interesting finding is discussed in the text, lines 246-269 and 375-383 (see also our answers on page 3 above and page 7 below).

Major Comments

Evidence and Claims:

* *While the methodological aspect is very strong the balance between presenting a novel method and presenting specific cell biological findings needs to be improved. Now it is quite unclear what the manuscript wants to present.*

* *The abstract needs a complete overhaul. From reading the abstract, it is not clear what the manuscript wants to present.*

We have modified the abstract to make it more clear that we do not make any general claims on the impact of vimentin on the interactions and functions of different organelles, but rather describe what can be directly observed after the acute displacement of vimentin and which conclusions can be made from these observations.

Regarding the research findings there are a number of things for the authors to consider. Since the methods aspect is, in the eyes of this reviewer, in focus, I have not stringently assessed the experimental findings. Hence, the comments below are things to be considered in order to make the findings related to IF research stronger:

* *Cell-specific keratin interactions: The manuscript could benefit from some further validation of the physical interactions between vimentin and keratin-8 across different cell types.*

We have improved the images of keratin and vimentin by using super-resolution (Airyscan) microscopy to show that they indeed form distinct filaments in HeLa cells, whereas in COS-7 and U2OS cells, where their co-displacement occurs, they can also incorporate into the same filaments. This observation was very surprising but agrees with the data published by the Gelfand lab on similarity in the distribution pattern and co-transport of vimentin and keratin in RPE1 cells (Robert et al, 2019, FASEB J). Colocalization and coprecipitation of vimentin and keratin has been also described by Velez-delValle et al. in epithelial cells (Sci Rep 2016).

* *Impact on microtubules: The disorganization of stable microtubules in cells expressing KIF5A was attributed to overexpression effects. It would be helpful to include additional controls, such as expressing KIF5A without vimentin constructs, to confirm this claim.*

This control has been included in the new Figure S3. We note that this observation fully aligns with data published by another lab (Andreu-Carbó et al, 2024, Nat Comm).

* *ER-vimentin linkages: The observation that ER-vimentin interactions persist in RNF26 knockout cells is intriguing. The manuscript would benefit from a discussion on possible candidates for alternative linkers.*

We have added a short discussion (lines 394-398) about the potential involvement of nesprins, such as nesprin-3, because they can connect the nuclear envelope to intermediate filaments, and might also partly participate in ER sheet-IF connections because ER and nuclear membranes are continuous and show some overlap in proteome.

* *Construct variability: Do the authors have some data on how much Expression level differences significantly affect the outcomes (e.g., incomplete recovery)?*

We have added a figure (Figure S2B), which shows that incomplete recovery of vimentin clustering does not correlate with protein expression levels and likely depends on other factors, which could possibly be the cell cycle phase or degree of vimentin entanglement after repositioning. This point is discussed in revised text, lines 194-197.

Reviewer #2 (Significance (Required)):

Significance

General Assessment: The study represents a significant technical advance in the study of cytoskeletal dynamics. The tools developed address critical limitations of traditional vimentin perturbation methods, allowing for spatiotemporally precise manipulation without long-term effects on gene expression or signaling pathways.

Novelty:

This is, to my knowledge, the first demonstration of reversible and acute vimentin repositioning using optogenetics. The study extends understanding of vimentin's short-term mechanical and organizational roles, distinguishing them from compensatory effects observed in knockdown models.

Audience and Impact: The manuscript will appeal to researchers in cytoskeletal dynamics, cell mechanics, and organelle biology. The tools have broader applicability in studying other cytoskeletal systems and could inspire translational applications, such as investigating the role of vimentin in cancer or fibrosis.

The reference list provide a relatively representative selection of articles relevant for the article. However, the authors may consider whether there could be relevant information in the relatively recent special edition of Current Opinion in Cell Biology, which focused on IFs, specially featuring vimentin <https://www.sciencedirect.com/special-issue/10TFHK2QCKW>

We thank the reviewer for this excellent suggestion, and we have included some additional references from this issue.

Field of Expertise

I specialize in cell biology, intermediate filaments, post-translational modifications, cytoskeletal dynamics, and advanced microscopy techniques.

****Referees cross-commenting****

This session contains comments of Reviewer 1 and Reviewer 2

Reviewer 1:

I don't understand what Reviewer 2 means by "A major shortcoming is the unclear narrative, what do the authors want to present? This aspect requires significant attention." I found the narrative, purpose and conclusions of this study very clear to me. I also do not understand Reviewer 2's concern with the abstract. I re-read it and it still seems very clear and appropriate to me. For example, the authors state "Here, we present tools that allow rapid manipulation of vimentin IFs in the whole cytoplasm or within specific subcellular regions by inducibly coupling them to microtubule motors, either pharmacologically or using light". This seems clear and correct to me. It would be helpful if Reviewer 2 could point to specific language and explain why it is problematic.

Reviewer 2:

The strength of this paper is clearly the strong methods development and I find this aspect very intriguing and attractive. There is an imbalance in the narrative presenting on one hand the method and on the other hand presenting concrete research results. In my view, although interesting, the different experimental results serve more as proof-of-concept and they should not be presented as bona fide evidence of an existing or lacking bilateral interrelationship. Indeed, the cited sentence makes sense: "Here, we present tools that allow rapid manipulation of vimentin IFs in the whole cytoplasm or within specific subcellular regions by inducibly coupling them to microtubule motors, either pharmacologically or using light." as it features the methods aspect of the paper. However, the following sentences: "Perinuclear clustering of vimentin had no strong effect on the actin or microtubule organization, cell spreading, and focal adhesions, but reduced cell stiffness. Mitochondria and endoplasmic reticulum sheets were repositioned together with vimentin, whereas lysosomes were only briefly repositioned and rapidly regained their normal distribution. Keratin was displaced along with vimentin in some cell lines but remained intact in others. " embraces everything from actin to microtubules to cell spreading to focal adhesions to cell stiffness to mitochondrial function to lysosomes to interactions with other IF family members etc. This gives the impression that the authors want to make claims on how vimentin affects or does not affect these cellular functions and structures and once just cannot make such sweeping claims with so little evidence. With the experimental setting included, non of these claims can be really made without rigidly examining each and every interaction (which has been done separately for many of these bilateral interactions during the past 20 years or so).

Hence, it should be made clear that these observations are used and mentioned as proof of concept that the tool is working, not as evidence that this or that interaction takes place or does

not take place. As I indicated in my review, such claims on any of these bilateral interactions would require a lot more evidence to be properly substantiated.

We would like to point out that we do not make any 'sweeping claims' about bilateral interactions – we just describe, based on careful quantification, which structures are acutely displaced or affected in the assays that we use. These data do not mean that vimentin does not interact at all with certain structures or cannot affect their distribution or function – it just means that acute perinuclear clustering of vimentin does not visibly perturb certain structures in the conditions and cell types that we use. We tweaked the abstract to emphasize further that we are looking at rapid, immediate effects of vimentin repositioning. The current version of the abstract literally describes our observations without making any broad conclusions about the functional interplay between vimentin and other cell structures.

My comment is to be regarded as a positive one. If I would judge the paper based on how one could interpret the abstract and the text regarding, for example, that vimentin does not affect focal adhesions but changes cellular stiffness, my review would be significantly more stringent. However, I would really like to see this paper being published, but the claims on revealing new vimentin functions or disproving earlier observations based on these very limited data are just not sufficiently substantiated to be acceptable.

We absolutely do not claim that vimentin does not affect focal adhesions – we report that perinuclear clustering of vimentin has no short-term effect on the number of focal adhesions in the cells we have studied. We are not disproving any previous data. To make this more clear, we have re-written the discussion to emphasize that our tools could lead to different effects in other cell types (e.g. rapidly migrating cells such as fibroblasts) and that our tools could be useful for probing for local effects of vimentin on cell polarity and migration.

Hence, I urge the authors to adjust the narrative to be clear on the methods development, which is also the focus of the title. I believe this is a justified recommendation and also, overall, a fair shake of the study and a constructive approach on how to publish this manuscript without extensive experiments.

We thank the reviewer for the constructive comment.

Reviewer 1:

I thank Reviewer 2 for this explanation. I do understand their point. However, while not the end of the story, I do feel the authors' data are a bit more than just a proof of principle and do offer important insights into the biology which the field will need to grapple with. Each graph includes measurements on dozens of cells from multiple experiments and there is clearly selectivity to what segregates with the vimentin filaments and what does not. I would just ask the authors to be a bit more nuanced in their interpretation and conclusions about the biology to address Reviewer 2's concerns.

We fully agree with this comment, and we have modified the paper to introduce more nuance in our interpretation, see lines 362-372 of the Discussion.

Reviewer 2:

Full Revision

That sounds like a fair assessment. Main thing is that this data is presented in a balanced way, with emphasis on the model development. Some of the presented data are in contradiction with quite established concepts by several researchers and the data presented here does not substantiate a paradigm shift. Regardless of this, some pieces of the data are intriguing, for example, the live cell imaging.

Reviewer #3 (Evidence, reproducibility and clarity (Required)):

Summary:

This is an excellent paper describing the use of chemical and light-induced heterodimerization of microtubule-based motors to rapidly disrupt the distribution of the vimentin cytoskeletal network. Rapid clustering of vimentin did not significantly affect the microtubule or actin networks, cell spreading or focal adhesions. Other organelles were repositioned together with vimentin. Interestingly, in some cell lines, keratin networks were displaced along with vimentin while in other cells they were not.

Major comments:

The conclusions are well supported by the data presented and appropriate controls are included.

Optional comments:

1. The authors should expand on why they think the plus end directed KIF5A gives such a strong localization of vimentin to the perinuclear area.

We think that two factors can contribute to this counterintuitive effect. First, vimentin is strongly concentrated and entangled in the perinuclear region, and displacement of some vimentin filaments to the cell periphery can cause the collapse of the rest to the cell center, with kinesins being unable to pull the perinuclear network apart. Second, kinesin-1 KIF5A is a motor that strongly prefers stable, post-translationally modified microtubules, and our previous study has shown that a significant proportion of such microtubules are located with their minus ends facing towards the cell periphery (Chen et al., Elife 2016). This could contribute to the accumulation of vimentin in the cell center upon KIF5A recruitment. These considerations were added to the revised text, lines 344-347.

2. Consideration should be given to the idea that the pulling of ER and mitochondria along with the vimentin could be due to trapping of these organelles within the vimentin matrix and not necessarily due to direct interactions. Such reasoning could explain the transient localization of lysosomes with the center aggregate since lysosomes are generally not thought to significantly bind to vimentin networks.

This is an excellent point, and we have included it in the revised article, lines 333-335 and 405.

Reviewer #3 (Significance (Required)):

This study describes some valuable tools that should be useful to cell biologists interested in determining the role of the cytoskeleton and possibly other organelles in a variety of cellular contexts. It overcomes some of the existing shortcomings of the pharmacological reagents currently available for studying intermediate filament biology and will provide a useful adjunct to other more long-term manipulations of the cytoskeleton. While much of the data presented confirm results obtained by other methods, this is a significant technical advance as it provides a short time scale, and in one instance, reversible manipulation of the cytoskeleton.

May 24, 2025

RE: JCB Manuscript #202504004T

Anna Akhmanova
Utrecht University

Dear Dr. Akhmanova,

Thank you for submitting your manuscript "Optogenetic and chemical genetic tools for rapid repositioning of vimentin intermediate filaments" to the Journal of Cell Biology. We are happy to let you know that we are ready to accept the paper, provided you address the minor points raised by Reviewer #3 as well as final revisions necessary to meet our formatting guidelines (see details below). These are all quite manageable, and once revised, the paper will be in great shape for publication.

A. MANUSCRIPT ORGANIZATION AND FORMATTING:

1) Text limits: Character count for Tools is < 40,000, not including spaces. Count includes title page, abstract, introduction, results, discussion, and acknowledgments. Count does not include materials and methods, figure legends, references, tables, or supplemental legends.

2) Figure formatting: Tools may have up to 10 main text figures. Scale bars must be present on all microscopy images, including inset magnifications. Molecular weight or nucleic acid size markers must be included on all gel electrophoresis. Please avoid pairing red and green for images and graphs to ensure legibility for color-blind readers. If red and green are paired for images, please ensure that the particular red and green hues used in micrographs are distinctive with any of the colorblind types. If not, please modify colors accordingly or provide separate images of the individual channels.

3) Statistical analysis: Error bars on graphic representations of numerical data must be clearly described in the figure legend. The number of independent data points (n) represented in a graph must be indicated in the legend. Please indicate whether 'n' refers to technical or biological replicates (i.e. number of analyzed cells, samples or animals, number of independent experiments). If independent experiments with multiple biological replicates have been performed, we recommend using distribution-reproducibility SuperPlots (please see Lord et al., JCB 2020) to better display the distribution of the entire dataset, and report statistics (such as means, error bars, and P values) that address the reproducibility of the findings.

Statistical methods should be explained in full in the materials and methods. For figures presenting pooled data the statistical measure should be defined in the figure legends. Please also be sure to indicate the statistical tests used in each of your experiments (both in the figure legend itself and in a separate methods section) as well as the parameters of the test (for example, if you ran a t-test, please indicate if it was one- or two-sided, etc.). Also, if you used parametric tests, please indicate if the data distribution was tested for normality (and if so, how). If not, you must state something to the effect that "Data distribution was assumed to be normal but this was not formally tested."

4) Materials and methods: Should be comprehensive and not simply reference a previous publication for details on how an experiment was performed. Please provide full descriptions (at least in brief) in the text for readers who may not have access to referenced manuscripts. The text should not refer to methods "...as previously described."

5) For all cell lines, vectors, strains, constructs/cDNAs, etc. - all genetic material: please include database / vendor ID (e.g. Addgene, ATCC, etc.) or if unavailable, please briefly describe their basic genetic features, even if described in other published work or gifted to you by other investigators (and provide references where appropriate). Please be sure to provide the sequences for all of your oligos: primers, si/shRNA, RNAi, gRNAs, etc. in the materials and methods. You must also indicate in the methods the source, species, and catalog numbers/vendor identifiers (where appropriate) for all of your antibodies, including secondary. If antibodies are not commercial, please add a reference citation if possible.

6) Microscope image acquisition: The following information must be provided about the acquisition and processing of images:

- Make and model of microscope
- Type, magnification, and numerical aperture of the objective lenses
- Temperature

- d. Imaging medium
- e. Fluorochromes
- f. Camera make and model
- g. Acquisition software
- h. Any software used for image processing subsequent to data acquisition. Please include details and types of operations involved (e.g., type of deconvolution, 3D reconstitutions, surface or volume rendering, gamma adjustments, etc.).

7) References: There is no limit to the number of references cited in a manuscript. References should be cited parenthetically in the text by author and year of publication. Abbreviate the names of journals according to PubMed.

8) Supplemental materials: Tools may have up to 5 supplemental figures and 10 videos. Tables, like figures, should be provided as individual, editable files. A summary of all supplemental material should appear at the end of the Materials and methods section. Please include one brief sentence per item. Unropped gels/blots should not be included as supplemental material but submitted separately as Source Data (see pt#14 below).

9) Video legends: Should describe what is being shown, the cell type or tissue being viewed (including relevant cell treatments, concentration and duration, or transfection), the imaging method (e.g., time-lapse epifluorescence microscopy), what each color represents, how often frames were collected, the frames/second display rate, and the number of any figure that has related video stills or images.

10) eTOC summary: A ~40-50 word summary that describes the context and significance of the findings for a general readership should be included on the title page. The statement should be written in the present tense and refer to the work in the third person. It should begin with "First author name(s) et al..." to match our preferred style.

11) Conflict of interest statement: JCB requires inclusion of a statement in the acknowledgements regarding competing financial interests. If no competing financial interests exist, please include the following statement: "The authors declare no competing financial interests." If competing interests are declared, please follow your statement of these competing interests with the following statement: "The authors declare no further competing financial interests."

12) A separate author contribution section is required following the Acknowledgments in all research manuscripts. All authors should be mentioned and designated by their first and middle initials and full surnames. We encourage use of the CRediT nomenclature (<https://casrai.org/credit/>).

13) ORCID IDs: ORCID IDs are unique identifiers allowing researchers to create a record of their various scholarly contributions in a single place. Please note that ORCID IDs are required for all authors. At resubmission of your final files, please be sure to provide your ORCID ID and those of all co-authors.

14) JCB requires authors to submit Source Data used to generate figures containing gels and Western blots with all revised manuscripts. This Source Data consists of fully uncropped and unprocessed images for each gel/blot displayed in the main and supplemental figures. For assays performed using capillary electrophoresis and/or immunoassay-based detection, authors should instead provide the electropherogram graph(s) for each experiment, plotting fluorescence/chemiluminescence intensity vs. molecular weight/size. Since your paper includes cropped gel and/or blot images, please be sure to provide one Source Data file for each figure gels, blots, and/or capillary electrophoresis assays along with your revised manuscript files. File names for Source Data figures should be alphanumeric without any spaces or special characters (i.e., SourceDataF#, where F# refers to the associated main figure number or SourceDataFS# for those associated with Supplementary figures). For traditional gels and blots, the lanes of the gels/blots should be labeled as they are in the associated figure, the place where cropping was applied should be marked (with a box), and molecular weight/size standards should be labeled wherever possible. For capillary electrophoresis assays, each trace in the graph should be color-coded and labeled to indicate which protein, gene, or sample is being measured (please try to avoid red/green combinations to accommodate our color-blind readers).

Source Data files will be directly linked to specific figures in the published article. Source Data Figures should be provided as individual PDF files (one file per figure). Authors should endeavor to retain a minimum resolution of 300 dpi or pixels per inch. Please review our instructions for export from Photoshop, Illustrator, and PowerPoint here: <https://rupress.org/jcb/pages/submission-guidelines#revised>

15) Journal of Cell Biology now requires a data availability statement for all research article submissions. These statements will be published in the article directly above the Acknowledgments. The statement should address all data underlying the research presented in the manuscript. Please visit the JCB instructions for authors for guidelines and examples of statements at (<https://rupress.org/jcb/pages/editorial-policies#data-availability-statement>).

B. FINAL FILES:

- An editable version of the final text (.DOC or .DOCX) is needed for copyediting (no PDFs).
- A document addressing the remaining reviewers comments. Please also highlight all changes in the text of the manuscript.
- High-resolution figure and MP4 video files: See our detailed guidelines for preparing your production-ready images, <https://jcb.rupress.org/fig-vid-guidelines>.
- Cover images: If you have any striking images related to this story, we would be happy to consider them for inclusion on the journal cover. Submitted images may also be chosen for highlighting on the journal table of contents or JCB homepage carousel. Images should be uploaded as TIFF or EPS files and must be at least 300 dpi resolution.

****It is JCB policy that if requested, original data images must be made available to the editors. Failure to provide original images upon request will result in unavoidable delays in publication. Please ensure that you have access to all original data images prior to final submission.****

****The license to publish form must be signed before your manuscript can be sent to production. A link to the electronic license to publish form will be sent to the corresponding author only. Please take a moment to check your funder requirements before choosing the appropriate license.****

Thank you for your attention to these final processing requirements. Please revise and format the manuscript and upload materials within 7 days. If you need an extension for whatever reason, please let us know and we can work with you to determine a suitable revision period.

Thank you for this interesting contribution, we look forward to receiving the final version soon and publishing your paper in Journal of Cell Biology.

All the best,

Sandrine Etienne-Manneville, PhD
Monitoring Editor
Journal of Cell Biology

Dan Simon, PhD
Scientific Editor
Journal of Cell Biology

Reviewer #1 (Comments to the Authors (Required)):

The authors have responded admirably to my critique, providing additional data that fully address my concerns. The data are of high quality and rigor and the technique is innovative. The authors' approach is an important contribution to the cytoskeletal toolbox and should be of broad interest to cell biologists who study cytoskeletal interactions, particularly the interaction of intermediate filaments with other cellular organelles or cytoskeletal polymers, or the role of intermediate filaments in cellular mechanics

Reviewer #2 (Comments to the Authors (Required)):

This paper describes optogenetic and pharmacological tools to clear the cytoplasm of vimentin by coupling the network to motor proteins. This represents a significant advance in the IF field and should be generally applicable to other IF proteins. They provide additional evidence of interactions between the vimentin and keratin cytoskeletons, potentially opening up new areas of investigation for which the described tools will be extremely helpful.

Reviewer #3 (Comments to the Authors (Required)):

This outstanding manuscript presents innovative optogenetic tools for acute, reversible, and spatially controlled manipulation of vimentin intermediate filaments (IFs). These tools fill a longstanding gap in the cytoskeletal toolkit, allowing researchers to probe the real-time role of vimentin in live cells with minimal disruption to other cellular components. In its own way, this is a landmark paper for cytoskeletal research. The methods are a genuine advance, and the biological insights-particularly regarding IF-organelle interactions and mechanical contributions-are meaningful. With minor edits and adjustments to tone and causality claims, the paper will be an important and widely used reference.

Key strengths:

Technical innovation: The dual-system approach (light- and chemically-induced dimerization) enables precise, fast, and reversible vimentin repositioning-without gene knockouts or non-specific drugs.

Comprehensive validation: The authors validate the tools across multiple cell types, quantify mechanical and organelle effects, and demonstrate minimal short-term impact on actin, microtubules, or cell viability.

Biological insight: Vimentin clustering reduces cell stiffness and alters ER and mitochondria positioning, providing mechanistic insights into cytoskeletal and organelle interplay.

Causality vs. correlation: The interpretation of the results is significantly improved from the first version. For the authors to consider how to best make sure that it is obvious to the reader when the observations made by the new methods are primarily correlative. For example, that the observed spatial repositioning represents a correlation with organelle relocalization, without assuming direct causality.

Generalizability: Also for the authors to consider, the cell-type specificity of keratin-vimentin interactions should be acknowledged; broader conclusions should be avoided.

Tool optimization: A note on expression level sensitivity and construct-specific side effects (e.g., KIF5A) should be included to guide reproducibility.

Minor suggestions for clarity:

Would there be a possibility to add quantitative summaries (e.g., expression histograms, phenotype tables).

Standardize construct nomenclature and video legends.

Consider a graphical summary for broader accessibility.

Conclusion:

This is a well-executed, impactful study that sets a new standard for IF research. I strongly support publication after minor revisions to tone down overstatements and enhance clarity.

Response to Reviewers

Reviewer #1:

The authors have responded admirably to my critique, providing additional data that fully address my concerns. The data are of high quality and rigor and the technique is innovative. The authors' approach is an important contribution to the cytoskeletal toolbox and should be of broad interest to cell biologists who study cytoskeletal interactions, particularly the interaction of intermediate filaments with other cellular organelles or cytoskeletal polymers, or the role of intermediate filaments in cellular mechanics.

We thank the Reviewer for the positive comments.

Reviewer #2:

This paper describes optogenetic and pharmacological tools to clear the cytoplasm of vimentin by coupling the network to motor proteins. This represents a significant advance in the IF field and should be generally applicable to other IF proteins. They provide additional evidence of interactions between the vimentin and keratin cytoskeletons, potentially opening up new areas of investigation for which the described tools will be extremely helpful.

We thank the Reviewer for the positive comments.

Reviewer #3:

This outstanding manuscript presents innovative optogenetic tools for acute, reversible, and spatially controlled manipulation of vimentin intermediate filaments (IFs). These tools fill a longstanding gap in the cytoskeletal toolkit, allowing researchers to probe the real-time role of vimentin in live cells with minimal disruption to other cellular components. In its own way, this is a landmark paper for cytoskeletal research. The methods are a genuine advance, and the biological insights-particularly regarding IF-organelle interactions and mechanical contributions-are meaningful. With minor edits and adjustments to tone and causality claims, the paper will be an important and widely used reference.

We thank the Reviewer for the positive comments.

Key strengths:

Technical innovation: The dual-system approach (light- and chemically-induced dimerization) enables precise, fast, and reversible vimentin repositioning-without gene knockouts or non-specific drugs.

Comprehensive validation: The authors validate the tools across multiple cell types, quantify mechanical and organelle effects, and demonstrate minimal short-term impact on actin, microtubules, or cell viability.

Biological insight: Vimentin clustering reduces cell stiffness and alters ER and mitochondria positioning, providing mechanistic insights into cytoskeletal and organelle interplay.

Causality vs. correlation: The interpretation of the results is significantly improved from the first version. For the authors to consider how to best make sure that it is obvious to the reader when the observations made by the new methods are primarily correlative. For example, that the observed spatial repositioning represents a correlation with organelle relocation, without assuming direct causality.

We do not make any conclusions about direct connections between vimentin and other cellular structures, although we do mention that some of our data are consistent with previously described direct connections of vimentin, with, e.g., ER and mitochondria. Since our perturbations are tightly time-controlled by drug or by light, it seems more logical not to talk about “correlation” or “causality” but about the effects of acute vimentin redistribution on the distribution of other cellular structures. Throughout the text, we describe what has been observed. We have adjusted some statements. For example, we now state that “we observed during vimentin pulling a concomitant retraction of the ER sheets and matrices”.

Generalizability: Also for the authors to consider, the cell-type specificity of keratin-vimentin interactions should be acknowledged; broader conclusions should be avoided.

This is exactly what we do. The corresponding section of the text is entitled “Co-dependency of keratin-8 and vimentin localization is cell-type dependent”, and in the Discussion, we state: “We identified cell-type-specific differences in the interactions between vimentin and keratin. In COS-7 and U2OS cells, where the two IF types are co-expressed (termed hybrid cell states by Sha et al., 2019) but keratin levels are relatively low, super-resolution microscopy indicated that keratin and vimentin can co-assemble into the same filaments, and keratin-8 was pulled along with vimentin. In HeLa cells, where keratin levels are higher, co-assembly seemed to be less frequent, and vimentin clustering did not impact the keratin network. Colocalization of vimentin and keratin in certain epithelial cells has been detected previously (Robert et al., 2019; Velez-delValle et al., 2016) but has not received much attention, and it deserves further investigation.”

Tool optimization: A note on expression level sensitivity and construct-specific side effects (e.g., KIF5A) should be included to guide reproducibility.

Construct expression levels used for analysis are quantified in Figures 1 and 2. The percentage of cells used for analysis is discussed in the text (e.g. lines 138-142 and 173-176), and the effect of KIF5A expression is discussed in the text (lines 225-226 and Figure S3).

Minor suggestions for clarity:

Would there be a possibility to add quantitative summaries (e.g., expression histograms, phenotype tables).

Expression levels are already quantified in Figure 1 and 2; adding phenotype tables would be complicated, and we do not think that this will add clarity compared to the current text.

Standardize construct nomenclature and video legends.

We have checked the construct nomenclature and corrected a couple of minor errors.

Consider a graphical summary for broader accessibility.

Graphical abstract has been added.

Conclusion:

This is a well-executed, impactful study that sets a new standard for IF research. I strongly support publication after minor revisions to tone down overstatements and enhance clarity.